# How Powerful is Implicit Denoising in Graph Neural Networks

## Abstract

Graph Neural Networks (GNNs), which aggregate features from neighbors, are widely used for graph-structured data processing due to their powerful representation learning capabilities. It is generally believed that GNNs can implicitly remove the non-predictive noises. However, the analysis of implicit denoising effect in graph neural networks remains open. In this work, we conduct a comprehensive theoretical study and analyze when and why the implicit denoising happens in GNNs. Specifically, we study the convergence properties of noise matrix. Our theoretical analysis suggests that the implicit denoising largely depends on the connectivity, the graph size, and GNN architectures. Moreover, we formally define and propose the adversarial graph signal denoising (AGSD) problem by extending graph signal denoising problem. By solving such a problem, we derive a robust graph convolution, where the smoothness of the node representations and the implicit denoising effect can be enhanced. Extensive empirical evaluations verify our theoretical analyses and the effectiveness of our proposed model.

## 1 Introduction

Graph Neural Networks (GNNs) (Kipf & Welling, 2017; Veličković et al., 2018; Hamilton et al., 2017) have been widely used in graph learning and achieved remarkable performance on graph-based tasks, such as traffic prediction (Guo et al., 2019), drug discovery (Dai et al., 2019), and recommendation system (Ying et al., 2018). A general principle behind Graph Neural Networks (GNNs) (Kipf & Welling, 2017; Veličković et al., 2018; Hamilton et al., 2017) is to perform a message passing operation that aggregates node features over neighborhoods, such that the smoothness of learned node representations on the graph is enhanced.

By promoting graph smoothness, the message passing and aggregation mechanism naturally leads to GNN models whose predictions are not only dependent on the feature of one specific node, but also the features from a set of neighboring nodes. Therefore, this mechanism can, to a certain extent, protect GNN models from noises: real-world graphs are usually noisy, e.g., Gaussian white noise exists on node features (Zhou et al., 2021), however, the influence of feature noises on the model's output could be counteracted by the feature aggregation operation in GNNs. We term this effect as *implicit denoising*.

While many works have been conducted in the empirical exploration of GNNs, relatively fewer advances have been achieved in theoretically studying this denoising effect. Early GNN models, such as the vanilla GCN (Kipf & Welling, 2017), GAT (Veličković et al., 2018) and GraphSAGE (Hamilton et al., 2017), propose different designs of aggregation functions, but the denoising effect is not discussed in these works. Some recent attempts (Ma et al., 2021b) are made to mathematically establish the connection between a variety of GNNs and the graph signal denoising problem (GSD) (Chen et al., 2014):

$$q(\mathbf{F}) = \min_{\mathbf{F}} \|\mathbf{F} - \mathbf{X}\|_F^2 + \lambda \operatorname{tr}\left(\mathbf{F}^\top \widetilde{\mathbf{L}} \mathbf{F}\right), \tag{1}$$

where $\mathbf{X} = \mathbf{X}^* + \boldsymbol{\eta}$ is the observed noisy feature matrix, $\boldsymbol{\eta} \in \mathbb{R}^{n \times d}$ is the noise matrix, $\mathbf{X}^*$ is the clean feature matrix, and $\widetilde{\mathbf{L}}$ is the graph Laplacian. The second term encourages the smoothness of the filtered feature matrix $\mathbf{F}$ over the graph., i.e., nearby vertices should have similar vertex features. By regarding the feature aggregation process in GNNs as solving a GSD problem, more advanced GNNs are proposed, such as GLP (Li et al., 2019), $S^2GC$ (Zhu & Koniusz, 2021), and IRLS (Yang et al.,

2021). Despite these prior attempts, little efforts have been made to rigorously study the denoising effect of message passing and aggregation operation. This urges us to think about a fundamental but not clearly answered question:

*Why and when implicit denoising happens in GNNs?*

In this work, we focus on the non-predictive stochasticity of noise in GNNs' aggregated features and analyze its properties. We prove that with the increase in graph size and graph connectivity factor, the stochasticity tends to diminish, which is called the "denoising effect" in our work. We will address this question using the tools from concentration inequalities and matrix theories, which are concerned with the study of the convergence of noise matrix. It offers a new framework to study the properties of graphs and GNNs in terms of the denoising effect. In order to facilitate our theoretical analysis, we derive Neumann Graph Convolution (NGC) from GSD. Specifically, to study the convergence rate, we introduce an insightful measurement on the convolution operator, termed *high-order graph connectivity factor*, which reveals how uniform the nodes are distributed in the neighborhood and reflects the strength of information diluted on a single neighboring node during the feature aggregation step. Intuitively, as the General Hoeffding Inequality (Hoeffding, 1994) (Lemma. D.1) suggests, a larger high-order graph connectivity factor, i.e., nodes are more uniformly distributed in the neighborhood, accelerates the convergence of the noise matrix and a larger graph size leads to faster convergence. Besides, GNN architectures also affect the convergence rate. Deeper GNNs can have a faster convergence rate.

To further strengthen the denoising effect, inspired by the adversarial training method (Madry et al., 2018), we propose the adversarial graph signal denoising problem (AGSD). By solving such a problem, we derive a robust graph convolution model based on the correlation of node feature and graph structure to increase the high-order graph connectivity factor, which helps us improve the denoising performance. Extensive experimental results on standard graph learning tasks verify our theoretical analyses and the effectiveness of our derived robust graph convolution model.

**Notations.** Let $\mathcal{G} = (\mathcal{V}, \mathcal{E})$ represent a undirected graph, where $\mathcal{V}$ is the set of vertices $\{v_1, \cdots, v_n\}$ with $|\mathcal{V}| = n$ and $\mathcal{E}$ is the set of edges. The adjacency matrix is defined as $\mathbf{A} \in \{0, 1\}^{n \times n}$, and $\mathbf{A}_{i,j} = 1$ if and only if $(v_i, v_j) \in \mathcal{E}$. Let $\mathcal{N}_i = \{v_j | \mathbf{A}_{i,j} = 1\}$ denote the neighborhood of node $v_i$ and $\mathbf{D}$ denote the diagonal degree matrix, where $\mathbf{D}_{i,i} = \sum_{j=1}^{n} \mathbf{A}_{i,j}$. The feature matrix is denoted as $\mathbf{X} \in \mathbb{R}^{n \times d}$ where each node $v_i$ is associated with a $d$-dimensional feature vector $\mathbf{X}_i$. $\mathbf{Y} \in \{0, 1\}^{n \times c}$ denotes the matrix, where $\mathbf{Y}_i \in \{0, 1\}^c$ is a one-hot vector and $\sum_{j=1}^{c} \mathbf{Y}_{i,j} = 1$ for any $v_i \in V$.

## 2 A Simple Unifying Framework: Neumann Graph Convolution

**A General Framework.** In this section, we discuss a simple yet general framework for solving graph signal denoising problem, namely Neumann Graph Convolution (NGC). Note that NGC is not a new GNN architecture. There also exist similar GNN architectures, such as GLP (Li et al., 2019), $S^2$GC (Zhu & Koniusz, 2021), and GaussianMRF (Jia & Benson, 2022). We focus on the theoretical analysis of the denoising effect in GNNs in this work. NGC can facilitate our theoretical analysis. By taking the derivative $\nabla q(\mathbf{F}) = 2\widetilde{\mathbf{L}}\mathbf{F} + 2(\mathbf{F} - \mathbf{X})$ to zero, we obtain the solution of GSD optimization problem as follows:

$$\mathbf{F} = (\mathbf{I} + \lambda \widetilde{\mathbf{L}})^{-1} \mathbf{X}. \tag{2}$$

To avoid the expensive computation of the inverse matrix, we can use Neumann series (Stewart, 1998) expansion to approximate Eq. (2) up to up to $S$-th order:

$$\left(\mathbf{I} + \lambda \widetilde{\mathbf{L}}\right)^{-1} = \frac{1}{\lambda + 1} \left(\mathbf{I} - \frac{\lambda}{\lambda + 1} \widetilde{\mathcal{A}}\right)^{-1} \approx \frac{1}{\lambda + 1} \sum_{s=0}^{S} \left(\frac{\lambda}{\lambda + 1} \widetilde{\mathcal{A}}\right)^s, \tag{3}$$

where $\widetilde{\mathcal{A}}$ can take the form of $\widetilde{\mathcal{A}} = \widetilde{\mathbf{D}}^{-\frac{1}{2}} \widetilde{\mathbf{A}} \widetilde{\mathbf{D}}^{-\frac{1}{2}}$ or $\widetilde{\mathcal{A}} = \widetilde{\mathbf{D}}^{-1} \widetilde{\mathbf{A}}$, and the proof can be found in Appendix B. Based on the Neumann series expansion of the solution of GSD, we introduce a general graph convolution model – Neumann Graph Convolution defined as the following expansion:

$$\mathbf{H} = \widetilde{\mathcal{A}}_S \mathbf{X} \mathbf{W} = \frac{1}{\lambda + 1} \sum_{s=0}^{S} \left(\frac{\lambda}{\lambda + 1} \widetilde{\mathcal{A}}\right)^s \mathbf{X} \mathbf{W}, \tag{4}$$

where $\widetilde{\boldsymbol{\mathcal{A}}}_S = \frac{1}{\lambda+1}\sum_{s=0}^{S}\left(\frac{\lambda}{\lambda+1}\widetilde{\boldsymbol{\mathcal{A}}}\right)^s$ and $\mathbf{W}$ is the weight matrix. Our spectral convolution $\widetilde{\boldsymbol{\mathcal{A}}}_S\mathbf{X}$ on graphs is a multi-scale graph convolution (Abu-El-Haija et al., 2020; Liao et al., 2019), which covers the single-scale graph convolution models such as SGC (Wu et al., 2019a) since the graph convolution of SGC is $\widetilde{\boldsymbol{\mathcal{A}}}^2\mathbf{X}$ and $\widetilde{\boldsymbol{\mathcal{A}}}^2$ is the third term of $\widetilde{\boldsymbol{\mathcal{A}}}_S$. Specifically, the N-GCN (Abu-El-Haija et al., 2020) extends the feature aggregation module in GCN by concatenating neighboring feature information using $\widetilde{\mathbf{A}}^k$ at each layer, while for NGC, we incorporate such multi-scale information by summing up $\left(\frac{\lambda}{\lambda+1}\widetilde{\boldsymbol{\mathcal{A}}}\right)^s$ as can be seen from Eq. (4). Besides, if we remove the non-linear functions in GCN (Kipf & Welling, 2017), it also can be covered by our model. Therefore, we can draw the conclusion that our proposed NGC is a general framework.

**High-order Graph Connectivity Factor.** Based on NGC, we obtain the filtered graph signal via $\mathbf{F} = \widetilde{\boldsymbol{\mathcal{A}}}_S\mathbf{X}$. Intuitively, $\widetilde{\boldsymbol{\mathcal{A}}}_S$ captures not only the connectivity of the graph structure (represented by $\widetilde{\boldsymbol{\mathcal{A}}}$), but also the higher order connectivity (represented by $\widetilde{\boldsymbol{\mathcal{A}}}^2, \widetilde{\boldsymbol{\mathcal{A}}}^3, \ldots, \widetilde{\boldsymbol{\mathcal{A}}}^S$). As will be discussed in Sec. 3, larger high order graph connectivity can accelerate the convergence of the noise feature matrix. To formally quantify the high order graph connectivity, we give the following definition:

**Definition 1** (High-order Graph Connectivity Factor). *We define the high-order graph connectivity factor $\tau$ as*

$$\tau = \max_{i}\tau_i, \quad \text{where } \tau_i = n\sum_{j=1}^{n}\left[\widetilde{\boldsymbol{\mathcal{A}}}_S\right]_{ij}^2 \bigg/ \left(1 - \left(\frac{\lambda}{\lambda+1}\right)^{S+1}\right)^2. \tag{5}$$

**Remark 1.** *Here we give some intuitions about why Eq. (5) represents high-order graph connectivity. Note that each element in $\widetilde{\boldsymbol{\mathcal{A}}}_S$ is non-negative and each row sum satisfies[1]*

$$\sum_{j=1}^{n}\left[\widetilde{\boldsymbol{\mathcal{A}}}_S\right]_{ij} = 1 - \left(\frac{\lambda}{\lambda+1}\right)^{S+1}. \tag{6}$$

*Based on Eq. (6), the sum of squares of elements in each row satisfy:*

$$\left(1 - \left(\frac{\lambda}{\lambda+1}\right)^{S+1}\right)^2 \bigg/ n \leq \sum_{j=1}^{n}\left[\widetilde{\boldsymbol{\mathcal{A}}}_S\right]_{ij}^2 \leq \left(1 - \left(\frac{\lambda}{\lambda+1}\right)^{S+1}\right)^2. \tag{7}$$

*When the high-order graph has a high connectivity, i.e., the elements in row $i$ of $\widetilde{\boldsymbol{\mathcal{A}}}_S$ are more uniformly distributed, Eq. (7) reaches its lower bound. Meanwhile, if the graph is not connected and there is only one element whose value is larger than $0$ in row $i$, Eq. (7) reaches its upper bound. Therefore, the value of $\tau \in [1, n]$ is determined as follows: when the high-order graph connectivity is high, $\tau \to 1$ and when the graph is less connected, $\tau \to n$.*

## 3 Main Theory

In this section, we analyze the denoising effect of NGC. Before we present our main theory, we first present our aggregation on noisy feature matrix and formulate four assumptions, which are necessary to construct our theory.

For the convenience of theoretical analysis, we adopt MSE loss[2] for our main theory. Consider $\widetilde{\boldsymbol{\mathcal{A}}}_S$ as our aggregation scheme, the NGC training based on Eq. (4) can be formulated as

$$\min_{\mathbf{W}} f(\mathbf{W}) = \left\|\widetilde{\boldsymbol{\mathcal{A}}}_S\mathbf{X}\mathbf{W} - \mathbf{Y}\right\|_F^2 = \left\|\widetilde{\boldsymbol{\mathcal{A}}}_S(\mathbf{X}^* + \boldsymbol{\eta})\mathbf{W} - \mathbf{Y}\right\|_F^2, \tag{8}$$

where $\mathbf{X}^*$ is the clean feature matrix, $\boldsymbol{\eta}$ denotes the noise added on $\mathbf{X}^*$, and $\mathbf{X} = \mathbf{X}^* + \boldsymbol{\eta}$ is the observed data matrix. Intuitively, if $\widetilde{\boldsymbol{\mathcal{A}}}_S\boldsymbol{\eta}$ is small enough, the added noise will not change the optimization direction on which the parameter is updated under the clean feature matrix $\mathbf{X}^*$. Before we present our main theory, we give four assumptions about noise $\boldsymbol{\eta}$, $\widetilde{\boldsymbol{\mathcal{A}}}_S$, and parameters $\mathbf{W}$.

---

[1]Note that this result is obtained by using $\widetilde{\boldsymbol{\mathcal{A}}} = \widetilde{\mathbf{D}}^{-1}\widetilde{\mathbf{A}}$ for the ease of theoretical analysis while in experiments we adopt more commonly used $\widetilde{\boldsymbol{\mathcal{A}}} = \widetilde{\mathbf{D}}^{-\frac{1}{2}}\widetilde{\mathbf{A}}\widetilde{\mathbf{D}}^{-\frac{1}{2}}$. The proof can be found in Appendix C.

[2]We consider MSE loss since it gives easier form of gradient and it can be extended to other losses satisfying certain conditions.

**Assumption 1.** *Each entry of the noise matrix $\boldsymbol{\eta}$, i.e., $[\boldsymbol{\eta}]_{ij}$ is i.i.d sub-Gaussian random variable with variance $\sigma$ and mean $\mu = 0$, i.e.,*

$$\mathbb{E}\left[e^{\lambda([\boldsymbol{\eta}]_{ij}-\mu)}\right] \leq e^{\sigma^2\lambda^2/2} \quad \text{for all } \lambda \in \mathbb{R}. \tag{9}$$

Note that it is common to assume that the noise follows Gaussian distribution (Zhou et al., 2021; Chen et al., 2021; Zhang et al., 2022), which is also covered by our sub-Gaussian assumption.

**Assumption 2.** *The high-order graph connectivity factor $\tau$ is $\mathcal{O}(n)$, i.e., $\lim_{n\to\infty}\frac{\tau}{n} = 0$.*

As we have discussed in Sec. 2, $\tau$ depends on the graph structure. In a well-connected graph, $\tau$ is usually relatively small compared with $n$. Only if all nodes of a graph are isolated, $\tau$ reaches its upper bound $n$.

**Assumption 3.** *The Frobenius norm of the parameter matrix $\mathbf{W}$ is bounded by a constant. There exists $C > 0$ such that $\|\mathbf{W}\|_F \leq C$, which is unrelated to $n$.*

We assume that the Frobenius norm of $\mathbf{W}$ is bounded by a constant. This is reasonable since recent advances in Neural Tangent Kernel (Jacot et al., 2018) indicate that over-parameterized network weights lie in the neighborhood of the small random initialization, which justifies Assumption 3.

**Assumption 4.** *The loss function in Eq. (8) is $L$-smooth,*

$$\|\nabla f(\mathbf{W}_1) - \nabla f(\mathbf{W}_2)\|_2 \leq L\|\mathbf{W}_1 - \mathbf{W}_2\|_2 \quad \text{for all } \mathbf{W}_1, \mathbf{W}_2 \in \mathbb{R}^{d\times c}. \tag{10}$$

The $L$-smoothness of $f$ depends on the largest singular value of $\widetilde{\boldsymbol{\mathcal{A}}}_S \mathbf{X}$. For conciseness, we start with the smooth case. As the core part of our proof, we first derive the upper bound of the Frobenius norm of $\widetilde{\boldsymbol{\mathcal{A}}}_S \boldsymbol{\eta}$.

**Lemma 1.** *Suppose we choose $t = 2\tau \left(1 - \left(\frac{\lambda}{\lambda+1}\right)^S\right)^2 (4\log n + \log 2d)/n$. Then under Assumptions 1 and 2, with a high probability $1 - 1/d$, we have*

$$\left\|\widetilde{\boldsymbol{\mathcal{A}}}_S \boldsymbol{\eta}\right\|_F^2 \leq \frac{2\tau\left(1 - \left(\frac{\lambda}{\lambda+1}\right)^{S+1}\right)^2 \sigma^2 (4\log n + \log 2d)}{n}, \tag{11}$$

*where the proof can be found in Appendix D.*

Lemma 1 implies that the norm of the aggregated noise matrix $\widetilde{\boldsymbol{\mathcal{A}}}_S \boldsymbol{\eta}$ is bounded by three terms: the number of nodes of a graph $n$, the expansion order $S$, the high-order graph connectivity factor $\tau$. Intuitively, as the concentration bounds suggest, if we extract enough samples from the same sub-Gaussian variable, the average of these samples will converge to zero with a high probability. This requires our graph to be large enough and the sum of squares of the elements in the row of $\widetilde{\boldsymbol{\mathcal{A}}}_S$ to be small enough, which depends on the graph structure.

Now we start to present our main theorem for graph denoising. In order to demonstrate the effect of graph denoising, we further consider another loss function $g(\cdot)$ with the clean feature matrix:

$$g(\mathbf{W}) = \left\|\widetilde{\boldsymbol{\mathcal{A}}}_S \mathbf{X}^* \mathbf{W} - \mathbf{Y}\right\|_F^2. \tag{12}$$

Let $\mathbf{W}_g^* = \arg\min_{\mathbf{W}} g(\mathbf{W})$ be the minimizer of clean loss $g$, we aim to demonstrate that the learned model (from gradient descent on the noisy data $\mathbf{X}$) has essentially the same performance as $\mathbf{W}_g^*$ which is the optimal solution for the clean loss $g$.

**Theorem 1.** *Under Assumptions 1, 2, 3, 4 and Lemma 1, let $\mathbf{W}_f^{(k)}$ denote the $k$-th step gradient descent solution for $\min_{\mathbf{W}} f(\mathbf{W})$ with step size $\alpha \leq 1/L$, with probability $1 - 1/d$ we have*

$$g\left(\mathbf{W}_f^{(k)}\right) - g\left(\mathbf{W}_g^*\right) \leq \mathcal{O}\left(\frac{1}{2k\alpha}\right) + \mathcal{O}\left(\frac{\tau\log n}{n}\right), \tag{13}$$

*where $\mathbf{W}_g^* = \arg\min_{\mathbf{W}} g(\mathbf{W})$ is the optimal solution of the clean loss function $g(\mathbf{W})$, $\tau$ is the high-order graph connectivity factor, and $n$ is the number of nodes of a graph.*

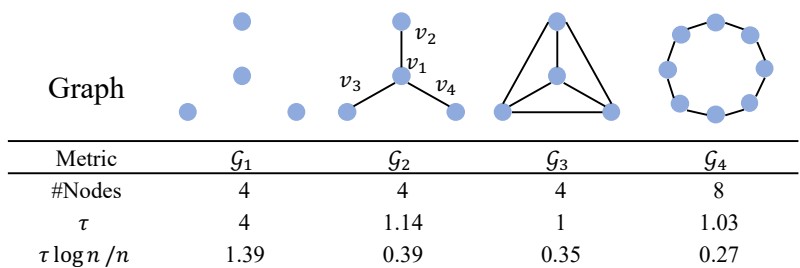

| Metric | $\mathcal{G}_1$ | $\mathcal{G}_2$ | $\mathcal{G}_3$ | $\mathcal{G}_4$ |
|---|---|---|---|---|
| #Nodes | 4 | 4 | 4 | 8 |
| $\tau$ | 4 | 1.14 | 1 | 1.03 |
| $\tau \log n /n$ | 1.39 | 0.39 | 0.35 | 0.27 |

Figure 1: An illustration of the graph structures on the implicit denoising performances. $\mathcal{G}_1$: nodes are isolated; $\mathcal{G}_2$: a star graph with 4 nodes; $\mathcal{G}_3$: a complete graph with 4 nodes; $\mathcal{G}_4$: a ring graph with 8 nodes. For computing $\tau$, $\lambda$ and $S$ are set to be 64.

The proof of Theorem 1 can be found in Appendix E.

**Remark 2.** *Denoising Effect. Theorem 1 suggests that the $k$-th step gradient descent solution $\mathbf{W}_f^{(k)}$ which is trained using the noisy feature matrix $\mathbf{X}$ enjoys a similar performance as the actual clean loss minimizer $\mathbf{W}_g^*$ with large enough $k$ and $n$. This implies the denoising effect of our proposed solution in Eq. (4).*

**Remark 3.** *Effect of graph structure on denoising. Note that the second term in Eq. (13) suggests that the denoising effect is linear with respect to $\tau$, which is directly related to the dataset graph structure. Specifically, as will be shown in Sec. 3, graph structure affects the value of $\tau$, and thus the denoising effect. A large well-connected graph tend to have a better denoising performance since $n$ is large and $\tau$ is close to 1.*

**Case Study: the Influence of Graph Structure on Implicit Denoising.** In this case study, we give four illustration samples in Figure 1. $G_1$, $G_2$, and $G_3$ have the same number of nodes. But the nodes on $G_1$ are isolated. $G_2$ has only one connected component and has a center node $v_1$ on the graph. $G_3$ is a complete graph such that there is an edge between any two nodes. In addition, we give a larger illustration graph $G_4$ to understand the influence of graph size. From Figure 1, we can extract the following insights: 1) There is no denoising effect (the value of $\tau \log n/n$ is quite large) on $G_1$ since the nodes are isolated. 2) The complete graph $G_3$ has the best denoising effect among the graphs of the same size since the values of elements in each row are distributed uniformly, leading to the lower bound of $\tau$. 3) Although $G_2$ has only one connected component, there is a center node $v_1$ on the graph. The existence of the center node makes the value of elements in each row imbalanced, which means that $\tau$ tends to have a larger value compared with $G_3$. 4) The decentralized graph like $G_4$ also can get a smaller $\tau$. 5) In terms of graph size, the graph with a larger size has a better denoising effect.

Combining our Theorem 1 and case study, we conclude that the denoising effect is influenced by graph size $n$ and the high-order graph connectivity factor $\tau$, which reflects the graph structure.

## 4 ROBUST NEUMANN GRAPH CONVOLUTION

In this section, we propose a new graph signal denoising problem - adversarial graph signal denoising (AGSD) problem to improve the denoising performance by deriving a robust graph convolution model.

### 4.1 ADVERSARIAL GRAPH SIGNAL DENOISING PROBLEM

Note that the second term in the GSD problem (Eq. (1)) which controls the smoothness of the feature matrix over graphs, is related to both the graph Laplacian and the node features. Therefore, the slight changes in the graph Laplacian matrix could lead to an unstable denoising effect. Inspired by the recent studies in adversarial training (Madry et al., 2018), we formulate the adversarial graph signal denoising problem as a min-max optimization problem:

$$\min_{\mathbf{F}} \left[ \|\mathbf{F} - \mathbf{X}\|_F^2 + \lambda \cdot \max_{\mathbf{L}'} \operatorname{tr}\left(\mathbf{F}^\top \mathbf{L}' \mathbf{F}\right) \right] \quad \text{s.t.} \quad \left\|\mathbf{L}' - \widetilde{\mathbf{L}}\right\|_F \leq \varepsilon. \tag{14}$$

Intuitively, the inner maximization on the Laplacian $\mathbf{L}'$ generates perturbations on the graph structure[3], and enlarges the distance between the node representations of connected neighbors. Such maximization finds the worst case perturbations on the graph Laplacian that hinders the global smoothness of $\mathbf{F}$ over the graph. Therefore, by training on those worse case Laplacian perturbations, one could obtain a robust graph signal denoising solution. Ideally, through solving Eq. (14), the smoothness of the node representations as well as the implicit denoising effect can be enhanced.

## 4.2 MINIMIZATION OF THE OPTIMIZATION PROBLEM

The min-max formulation in Eq. (14) also makes the adversarial graph signal denoising problem much harder to solve. Fortunately, unlike adversarial training (Madry et al., 2017) where we need to first adopt PGD to solve the inner maximization problem before we solve the outer minimization problem, here inner maximization problem is simple and has a closed form solution. In other words, we do not need to add random perturbations on the graph structure at each training epoch and can find the largest perturbation which maximizes the inner adversarial loss function. Denote the perturbation as $\boldsymbol{\delta}$, and $\mathbf{L}' = \widetilde{\mathbf{L}} + \boldsymbol{\delta}$. Directly solving[4] the inner maximization problem, we get $\boldsymbol{\delta} = \varepsilon \nabla h(\boldsymbol{\delta}) = \frac{\varepsilon \mathbf{F}\mathbf{F}^\top}{\|\mathbf{F}\mathbf{F}^\top\|_F}$. Plugging this solution into Eq. (14), we can rewrite the outer optimization problem as follows:

$$\rho(\mathbf{F}) = \min_{\mathbf{F}} \left[ \|\mathbf{F} - \mathbf{X}\|_F^2 + \lambda \max \operatorname{tr}\left(\mathbf{F}^\top \widetilde{\mathbf{L}}\mathbf{F}\right) + \lambda\varepsilon \operatorname{tr}\frac{\mathbf{F}^\top\mathbf{F}\mathbf{F}^\top\mathbf{F}}{\|\mathbf{F}\mathbf{F}^\top\|_F} \right]. \tag{15}$$

Taking the gradient of $\rho(\mathbf{F})$ to zero, we get the solution of the outer optimization problem as follows:

$$\mathbf{F} = \left(\mathbf{I} + \lambda\widetilde{\mathbf{L}} + \lambda\varepsilon\frac{\mathbf{F}\mathbf{F}^\top}{\|\mathbf{F}\mathbf{F}^\top\|_F}\right)^{-1}\mathbf{X}. \tag{16}$$

Both sides of Eq. (16) contains $\mathbf{F}$, directly computing the solution is difficult. Note that in Eq. (14) we also require $\mathbf{F}$ to be close to $\mathbf{X}$, we can approximate Eq. (16) by replacing the $\mathbf{F}$ with $\mathbf{X}$ in the inverse matrix on the right hand side. With the Neumann series expansion of the inverse matrix, we get the final approximate solution as

$$\mathbf{H} \approx \frac{1}{\lambda+1}\sum_{s=0}^{S}\left[\frac{\lambda}{\lambda+1}\left(\widetilde{\boldsymbol{\mathcal{A}}} - \frac{\varepsilon\mathbf{X}\mathbf{X}^\top}{\|\mathbf{X}\mathbf{X}^\top\|_F}\right)\right]^s\mathbf{X}\mathbf{W}. \tag{17}$$

The difference between Eq. (17) and Eq. (4) is that there is one more term in Eq. (17) derived from solving the inner optimization problem of Eq. (14). Based on this, we proposed our robust Neumann graph convolution (RNGC).

**Scalability.** Although RNGC introduces extra computational burdens for large graphs due to the $\mathbf{X}\mathbf{X}^\top$ term, if the feature matrix is sparse, the extra computational effort is minimal as the $\mathbf{X}\mathbf{X}^\top$ term can also be sparse. For the scalability of RNGC on large graphs with dense feature matrix, we only compute the inner product of feature vectors $(\mathbf{X}_i, \mathbf{X}_{j|j\in\mathcal{N}_i})$ between adjacent neighbors like masked attention in GAT. Compared with NGC, the additional computation cost is $\mathcal{O}(|\mathcal{E}|)$.

## 5 EXPERIMENTS

In this section, we conduct a comprehensive empirical study to understand the influence of different factors on the denoising effect of various models. To quantify the denoising effect, we test the model accuracy on noisy data on various GNN architectures and MLP for standard node classification tasks, where the noisy data is synthesized by mixing Gaussian noise with the original feature matrix. We also synthesize noisy data by flipping individual feature with a small Bernoulli probability on three citation datasets with binary features.

### 5.1 DENOISING EFFECTIVENESS COMPARISON OF VARIOUS GNN MODELS

In this section, we compare the denoising effectiveness of different GNN models through their test accuracy by training on the noisy feature matrix with Gaussian noise.

---

[3]Here we do not need exact graph structure perturbations as in graph adversarial attacks (Zügner et al., 2018; Zügner & Günnemann, 2019a) but a virtual perturbation that could lead to small changes in the Laplacian.

[4]More details on how to solve the inner maximization problem can be found in Appendix A.

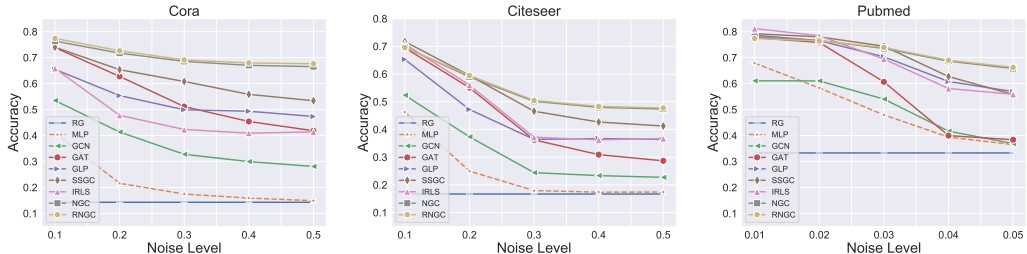

Figure 2: Comparison of classification accuracy v.s. noise level for semi-supervised node classification tasks. The noise level $\xi$ controls the magnitude of the Gaussian noise we add to the feature matrix: $\mathbf{X} + \xi\boldsymbol{\eta}$ where $\boldsymbol{\eta}$ is sampled from standard i.i.d., Gaussian distribution.

Table 1: Summary of results (10 runs) on heterophily graphs in terms of classification accuracy (%)

| Noise Level | Cornell | | Texas | | Wisconsin | | Actor | |
|---|---|---|---|---|---|---|---|---|
| | 0.01 | 1 | 0.01 | 1 | 0.01 | 1 | 0.01 | 1 |
| MLP | $69.7_{\pm8.6}$ | $55.3_{\pm7.6}$ | $69.7_{\pm8.6}$ | $55.3_{\pm7.6}$ | $\mathbf{78.6_{\pm6.5}}$ | $44.6_{\pm6.2}$ | $33.3_{\pm1.1}$ | $\mathbf{25.1_{\pm1.0}}$ |
| GCN | $56.9_{\pm8.4}$ | $51.7_{\pm14.9}$ | $56.6_{\pm8.1}$ | $52.5_{\pm11.7}$ | $48.0_{\pm6.1}$ | $41.2_{\pm9.0}$ | $26.4_{\pm1.0}$ | $23.8_{\pm3.0}$ |
| GAT | $55.8_{\pm8.9}$ | $55.0_{\pm7.5}$ | $56.4_{\pm8.1}$ | $54.7_{\pm8.2}$ | $53.4_{\pm7.2}$ | $\mathbf{48.2_{\pm7.2}}$ | $27.3_{\pm1.2}$ | $24.3_{\pm0.7}$ |
| GLP | $65.3_{\pm8.6}$ | $54.2_{\pm7.6}$ | $60.0_{\pm9.3}$ | $52.8_{\pm8.0}$ | $59.0_{\pm5.4}$ | $42.6_{\pm5.4}$ | $31.0_{\pm1.3}$ | $\mathbf{25.1_{\pm0.8}}$ |
| S$^2$GC | $60.6_{\pm9.3}$ | $48.6_{\pm10.4}$ | $56.4_{\pm7.2}$ | $50.3_{\pm8.0}$ | $47.4_{\pm4.5}$ | $37.2_{\pm3.7}$ | $27.2_{\pm1.1}$ | $23.5_{\pm1.3}$ |
| IRLS | $48.1_{\pm8.5}$ | $46.7_{\pm6.2}$ | $65.6_{\pm8.4}$ | $42.8_{\pm15.9}$ | $65.2_{\pm6.0}$ | $37.4_{\pm8.5}$ | $\mathbf{36.1_{\pm0.9}}$ | $21.5_{\pm4.1}$ |
| NGC | $\mathbf{72.8_{\pm8.7}}$ | $\mathbf{56.4_{\pm8.1}}$ | $\mathbf{73.9_{\pm6.9}}$ | $\mathbf{56.4_{\pm8.1}}$ | $74.8_{\pm6.8}$ | $\mathbf{46.8_{\pm6.6}}$ | $34.0_{\pm1.6}$ | $\mathbf{25.1_{\pm1.0}}$ |
| RNGC | $\mathbf{75.8_{\pm7.9}}$ | $\mathbf{56.4_{\pm8.1}}$ | $74.2_{\pm6.1}$ | $\mathbf{56.4_{\pm8.1}}$ | $76.4_{\pm5.3}$ | $\mathbf{46.8_{\pm6.6}}$ | $34.3_{\pm1.6}$ | $\mathbf{25.1_{\pm1.0}}$ |

**Datasets.** In our experiments, we utilize three public citation network datasets Cora, Citeseer, and Pubmed (Sen et al., 2008) which are homophily graphs for semi-supervised node classification. For the semi-supervised learning experimental setup, we follow the standard fixed splits employed in (Yang et al., 2016), with 20 nodes per class for training, 500 nodes for validation, and 1,000 nodes for testing. We also use four datasets: Cornell, Texas, Wisconsin, and Actor which are heterophily graphs for full-supervised node classification. For each dataset, we randomly split nodes into 60%, 20%, and 20% for training, validation, and testing as suggested in (Pei et al., 2020). Moreover, we utilize three large-scale graph datasets: Coauthor-CS, Coauthor-Phy (Shchur et al., 2018), and ogbn-products (Hu et al., 2020) for evaluation. For Coauthor datasets, we split nodes into 60%, 20%, and 20% for training, validation, and testing. For ogbn-products dataset, we follow the dataset split in OGB (Hu et al., 2020).

**Baselines.** For the baselines, we consider graph neural networks derived from graph signal denoising, including GLP (Li et al., 2019), S$^2$GC (Zhu & Koniusz, 2021), and IRLS (Yang et al., 2021); popular GNN architectures, such as GCN (Kipf & Welling, 2017) and GAT (Veličković et al., 2018); and MLP which has no aggregation operation.

**Experimental Setup and Implementations.** We assume that the original feature matrix is clean and do not have noise and we synthesize the noise from the standard Gaussian distribution and add them on the original feature matrix. By default, we apply row normalization for data after adding the Gaussian noise[5], and train all the models based on these noisy feature matrix. For the hyper-parameters of each model, we follow the setting that reported in their original papers. To eliminate the effect of randomness, we repeat such experiment for 100 or 10 times and report the mean accuracy. Note that in each repeated run, we add different Gaussian noises. While for the same run, we apply the same noisy feature matrix for training all the models. For our NGC and RNGC model, the hyper-parameter details can be found in Appendix H.2.

**Results on Supervised Node Classification.** Figure 2 illustrates the comparison of classification accuracy against the various noise levels for semi-supervised node classification tasks. The noise level $\xi$

---

[5]We also perform an analysis on the effect of row normalization in noisy feature matrix in Appendix I.1.

Table 2: Summary of results (10 runs) on Coauthor-CS and Coauthor-Phy in terms of accuracy (%)

| Noise Level | Coauthor-CS | | Coauthor-Phy | |
| --- | --- | --- | --- | --- |
| | 0.1 | 1 | 0.1 | 1 |
| MLP | $82.5_{\pm 1.8}$ | $22.3_{\pm 0.1}$ | $81.6_{\pm 8.1}$ | $47.0_{\pm 10.0}$ |
| GCN | $87.3_{\pm 0.5}$ | $61.3_{\pm 14.3}$ | $94.2_{\pm 0.4}$ | $78.6_{\pm 10.6}$ |
| GAT | $86.8_{\pm 3.6}$ | $57.9_{\pm 20.2}$ | $94.0_{\pm 0.4}$ | $63.7_{\pm 16.7}$ |
| GLP | $91.3_{\pm 0.4}$ | $52.4_{\pm 17.3}$ | $93.3_{\pm 2.5}$ | $81.3_{\pm 10.6}$ |
| $S^2$GC | $86.1_{\pm 0.2}$ | $79.6_{\pm 10.2}$ | $92.6_{\pm 1.3}$ | $89.4_{\pm 4.3}$ |
| IRLS | $78.8_{\pm 5.1}$ | $62.1_{\pm 17.8}$ | $89.2_{\pm 3.4}$ | $87.0_{\pm 4.5}$ |
| NGC | $\mathbf{95.3_{\pm 0.2}}$ | $\mathbf{87.1_{\pm 3.1}}$ | $\mathbf{95.7_{\pm 0.2}}$ | $\mathbf{93.1_{\pm 1.4}}$ |
| RNGC | $\mathbf{95.4_{\pm 0.2}}$ | $\mathbf{87.8_{\pm 1.5}}$ | $\mathbf{95.7_{\pm 0.2}}$ | $\mathbf{93.6_{\pm 0.8}}$ |

Table 3: Summary of results (10 runs) on ogbn-products in terms of accuracy (%)

| Noise Level | ogbn-products | |
| --- | --- | --- |
| | 0.1 | 1 |
| MLP | $59.68_{\pm 0.16}$ | $38.08_{\pm 0.10}$ |
| GCN | $75.60_{\pm 0.19}$ | $72.76_{\pm 0.20}$ |
| $S^2$GC | $74.95_{\pm 0.13}$ | $63.17_{\pm 0.12}$ |
| NGC | $\mathbf{77.56_{\pm 0.15}}$ | $\mathbf{73.36_{\pm 0.11}}$ |
| RNGC | $\mathbf{77.54_{\pm 0.15}}$ | $\mathbf{73.66_{\pm 0.13}}$ |

controls the magnitude of the Gaussian noise we add to the feature matrix: $\mathbf{X}+\xi\boldsymbol{\eta}$ where $\boldsymbol{\eta}$ is sampled from standard i.i.d., Gaussian distribution. For Cora and Citeseer, we test $\xi \in \{0.1, 0.2, 0.3, 0.4, 0.5\}$ and for Pubmed, we test $\xi \in \{0.01, 0.02, 0.03, 0.04, 0.05\}$. From Figure 2, we can observe that the test accuracy of MLP is close to randomly guessing (RG) when the noise level is relatively large. This implies the weak denoising effect of MLP models. For shallow GNN models, such as GCN and GAT (which usually contain 2 layers), their denoising performance is limited especially on Pubmed since they do not aggregate information (features and noise) from higher-order neighbors. For models with deep layers[6], such as IRLS ($\geq 8$ layers), the denoising performance is much better compared to shallow models. Lastly, our NGC and RNGC model with 16 layers ($S = 16$) achieve significantly better denoising performance compared with other baseline methods, which backup our theoretical analyses. In most cases, NGC and RNGC achieve very similar denoising performance but in general, RNGC still slightly outperforms NGC, suggesting that we indeed gain more benefits by solving the adversarial graph denoising problem.

On heterophilic graphs, MLPs are shown to achieve better performances in Table 1 than more GNNs such as GCN since feature aggregation may give inaccurate information. For our RNGC, note that we keep the zero-order term (no aggregation like MLP) with the largest weight and thus preserve the original feature information while also considering high-order terms for better denoising performances. Thus in the context of the denoising problem, the design of RNGC actually take the advantage of both zeroth-order and high-order information to achieve better denoising performances.

For ogbn-products, we only choose MLP, GCN, and $S^2$GC as baselines, since the results are sensitive concerning model size and various tricks from the OGB leaderboard. For fair comparison, the size of parameters for these baselines and RNGC is the same. We also use full-batch training for the baselines and our model. Table 2 and 3 report the comparison of classification accuracy against the various noise levels for full-supervised node classification tasks on large-scale graphs. The first- and second-highest accuracies are highlighted in bold. For these datasets, we test $\xi \in \{0.1, 1\}$. Compared with the above small datasets, the node degree on these three datasets is larger, which means they have better connectivity. From Table 2 and 3, we can observe that the test accuracy of MLP is far lower than GCN and RNGC. This implies the weak denoising effect of MLP. The test accuracy of GCN is slightly smaller than RNGC on these datasets since they are well-connected and have a large graph size and we can achieve a good denoising performance with shallow-layer GNN models. For the scalability of RNGC on large graphs such as ogbn-products, we use the acceleration method mentioned in Sec. 4.2.

## 5.2 DENOISING PERFORMANCE ON FEATURE FLIPPING PERTURBATION

In this section, we compare the denoising effectiveness of different models through their test accuracy by training on the noisy feature matrix which is perturbated through flipping the individual feature with a small Bernoulli probability on three citation datasets.

**Setting and Results.** We flip the individual feature on three citation datasets: Cora, Citeseer, and Pubmed as the noise. And we compare the denoising performance of RNGC with MLP and GCN. From Table 4 (the std information can be found in Appendix I.2), we can observe that the denoising

---

[6]We also perform an analysis on the denoising effect of depth in NGC and RNGC in Appendix I.2.

Table 4: Denoising performance over 100 runs against flipping perturbation

| Flipping probability | Cora | | | Citeseer | | | Pubmed | | |
|---|---|---|---|---|---|---|---|---|---|
| | 0.1 | 0.2 | 0.4 | 0.1 | 0.2 | 0.4 | 0.1 | 0.2 | 0.4 |
| MLP | 21.2 | 21.1 | 23.3 | 19.3 | 18.9 | 18.9 | 38.0 | 39.0 | 40.6 |
| GCN | 22.9 | 19.0 | 19.0 | 18.6 | 18.6 | 18.5 | 37.8 | 38.1 | 37.6 |
| GAT | 70.1 | 65.6 | 60.0 | 45.3 | 39.3 | 26.0 | 43.3 | 49.5 | 60.0 |
| GLP | 32.3 | 30.8 | 29.0 | 19.7 | 18.9 | 18.8 | 42.1 | 41.5 | 40.7 |
| $S^2GC$ | 75.0 | 71.5 | 63.8 | 49.9 | 46.4 | 43.4 | 50.4 | 60.2 | 69.3 |
| IRLS | 66.4 | 61.0 | 54.7 | 50.3 | 45.9 | 43.8 | 51.4 | 60.0 | 69.0 |
| NGC | **77.5** | **75.3** | 65.7 | 54.9 | 51.9 | 48.5 | 53.0 | 62.3 | 70.4 |
| RNGC | **77.6** | 75.2 | **72.8** | **55.0** | 51.8 | **48.7** | **54.3** | **63.9** | **71.6** |

Table 5: Defense performance over 100 runs against structure attack

| Model | Cora | Citeseer | Pubmed |
|---|---|---|---|
| GCN | 47.53 | 56.94 | 75.50 |
| GAT | 54.78 | 61.85 | 65.41 |
| RobustGCN | 50.51 | 55.35 | 67.95 |
| GCN-Jaccard | 60.82 | 59.89 | 83.66 |
| GCN-SVD | 52.06 | 57.18 | 82.72 |
| $S^2GC$ | 51.60 | 54.11 | 64.04 |
| RNGC | 63.16 | 65.64 | 84.04 |
| ProGNN | **69.72** | **68.95** | **86.76** |

performance of RNGC is much better than baselines when the flip probability is 0.4. In fact, the added perturbations by flipping the individual feature approximately follow a Bernoulli distribution, which is also a Sub-Gaussian distribution. The results verify our theoretical analysis further.

### 5.3 DEFENSE PERFORMANCE OF RNGC AGAINST GRAPH STRUCTURE ATTACK

Although we do not perform actual graph structure perturbations as in graph adversarial attacks (Zügner et al., 2018; Zügner & Günnemann, 2019a) but a virtual perturbation in the Laplacian. Therefore, it's not clear how much perturbations on the Laplacian correspond to the actual perturbations on graph structure. Nevertheless, we still conduct the experiments of RNGC against graph structure meta-attack where the ptb rate is 25%. As shown in the Table 5 (the std information can be found in Appendix I.2), our RNGC model still outperforms than GCN, GAT, RobustGCN (Zügner & Günnemann, 2019b), GCN-Jaccard (Wu et al., 2019b), GCN-SVD (Entezari et al., 2020), and $S^2GC$ on Cora, Citeseer, and Pubmed.

## 6 RELATED WORK

**Graph Signal Denoising** Existing graph denoising works are mainly based on the graph smoothing technique (Chen et al., 2014; Zhou et al., 2021). It is well known that GNNs can increase the smoothness of node features through aggregating information from neighbors, thus the influence from noisy features can be counteracted in GNN's output. Some GNN models are derived from the perspective of signal denoising, such as $S^2GC$ (Zhu & Koniusz, 2021), GLP (Li et al., 2019), and IRLS (Yang et al., 2021). Moreover, Ma et al. (2021b) builds the connection between signal denoising and existing GNNs by formulating message passing as a process of solving the GSD problem. This suggests a possibility for us to understand the behavior of GNNs through the lens of signal denoising. Besides, some works (Liu et al., 2021a; Fan et al., 2022; Ma et al., 2021a; Jin et al., 2021; Liu et al., 2021b; Jin et al., 2020; Zhang et al., 2022) have been proposed to conduct statistical analysis on the graph noise from the empirical perspective. In this work, we perform an extensive analysis to understand the denoising effect of GNNs from both theoretical and experimental perspectives.

**Smoothing and Over-smoothing.** One key principle of GNNs is to improve the smoothness of node representations. But stacking graph layers can lead to over-smoothing (Li et al., 2018), where the node representations can not be distinguishable. There are some recent works that have been proposed to address over-smoothing such as JKnet (Xu et al., 2018), GCNII (Chen et al., 2020), and RevGNN-Deep (Li et al., 2021). They add the output of shallow layers to the final layers with a residual-style design. In this work, we will show smoothing can help the denoising effect of GNNs.

## 7 CONCLUSION

Our work conducts a comprehensive study on the implicit denoising effect of graph neural networks. We theoretical show that the denoising effect of GNNs are largely influenced by the connectivity and the size of the graph structure, as well as the GNN architectures. Motivated by our analysis, we also propose a robust graph convolution model by solving the robust graph signal denoising problem which enhances the smoothness of node representations and the implicit denoising effect.

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

## A    THE DETAILS ON HOW TO SOLVE THE INNER MAXIMIZATION PROBLEM IN SEC. 4.2

Different from the non-concave inner maximization problem in the adversarial attack, our inner maximization problem is indeed a convex optimization problem. Hence, we do not need to add random perturbations on the graph structure at each training epoch and can find the largest perturbation which maximizes the inner adversarial loss function. Denote the perturbation as $\boldsymbol{\delta}$, and $\mathbf{L}' = \widetilde{\mathbf{L}} + \boldsymbol{\delta}$. We can rewrite the inner maximization problem as

$$\max_{\mathbf{L}'} \operatorname{tr}\left(\mathbf{F}^\top \mathbf{L}' \mathbf{F}\right) = \langle \widetilde{\mathbf{L}}, \mathbf{F}^\top \mathbf{F} \rangle + \max_{\boldsymbol{\delta}} \langle \boldsymbol{\delta}, \mathbf{F}^\top \mathbf{F} \rangle \quad \text{s.t.} \quad \|\boldsymbol{\delta}\|_F \leq \varepsilon. \tag{18}$$

We denote $h(\boldsymbol{\delta}) = \langle \boldsymbol{\delta}, \mathbf{F}^\top \mathbf{F} \rangle$. Obviously, $h(\boldsymbol{\delta})$ reaches the largest value when $\boldsymbol{\delta}$ has the same direction with the gradient of $h(\boldsymbol{\delta})$, e.g. $\boldsymbol{\delta} = \varepsilon \nabla h(\boldsymbol{\delta}) = \frac{\varepsilon \mathbf{F} \mathbf{F}^\top}{\|\mathbf{F} \mathbf{F}^\top\|_F}$, which is illustrated in Fig. 3.

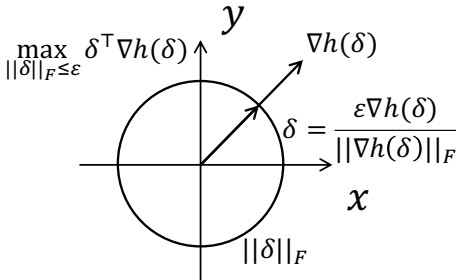

Figure 3: The illustration of the inner maximization problem. The adversarial loss function reaches the largest value when the direction of $\boldsymbol{\delta}$ is the same with $\nabla h(\boldsymbol{\delta})$

## B    ADDITIONAL DETAILS ON THE NEUMANN SERIES

We provide additional details and derivations on how to obtain the Neumann Series which leads to our Neumann Graph Convolution (NGC) method. Before we derive the Neumann Series, we first introduce the following lemmas which are crucial to the derivation of the Neumann Series.

**Lemma B.1.  (Gelfand formula) (Bhatia, 2013)**  *Given any matrix norm $\|\| \cdot \|\|$, then $\rho(\mathbf{A}) = \lim_{k \to \infty} \|\|\mathbf{A}^k\|\|^{1/k} = \inf_{k \geq 1} \|\|\mathbf{A}^k\|\|^{1/k} \leq \|\|\mathbf{A}\|\|.$*

Lemma B.1 describes the relationship between the spectral radius of a matrix and its matrix norm, *i.e.* $\rho(\mathbf{A}) = \lim_{k \to \infty} \|\|\mathbf{A}^k\|\|^{1/k}$.

**Lemma B.2.**  *Let $\mathbf{A} \in \mathbb{C}^{n \times n}$, the spectral radius $\rho(\mathbf{A}) = \max(\operatorname{abs}(\operatorname{spec}(\mathbf{A})))$, if $\rho(\mathbf{A}) < 1$, then $\sum_{k=0}^{\infty} \mathbf{A}^k$ converges to $(\mathbf{I} - \mathbf{A})^{-1}$.*

*Proof.* We first prove that $(\mathbf{I} - \mathbf{A})^{-1}$ exists as follows: Based on the definition of eigenvalues of $\mathbf{A}$, we have $|\lambda \mathbf{I} - \mathbf{A}| = 0$ and the solution is the eigenvalue of $\mathbf{A}$. Since $\rho(\mathbf{A}) < 1$, if $\lambda \geq 1$, then $|\lambda \mathbf{I} - \mathbf{A}| \neq 0$, so $|\mathbf{I} - \mathbf{A}| \neq 0$, which means $(\mathbf{I} - \mathbf{A})^{-1}$ exists.

Since $\rho(\mathbf{A}) < 1$ and by Lemma B.1, we have $\lim_{k \to \infty} \|\|\mathbf{A}^k\|\| = \rho(\mathbf{A})^k = 0$. Let $\mathbf{S}_k = \mathbf{A}^0 + \mathbf{A}^1 + \cdots + \mathbf{A}^k$, then we have

$$\lim_{k \to \infty} (\mathbf{S}^k - \mathbf{A}\mathbf{S}^k) = \lim_{k \to \infty} (\mathbf{I} - \mathbf{A})\mathbf{S}^k$$
$$= \lim_{k \to \infty} (\mathbf{I} - \mathbf{A}^{k+1})$$
$$= \mathbf{I}$$

Since $(\mathbf{I} - \mathbf{A})^{-1}$ exists, so we have $(\mathbf{I} - \mathbf{A}) \lim_{k \to \infty} \mathbf{S}^k = \mathbf{I}$, and $\lim_{k \to \infty} \mathbf{S}^k = (\mathbf{I} - \mathbf{A})^{-1}$, which finishes the proof.  □

Lemma B.2 describes the convergence of Neumann Series and the condition to get the convergence.

**Lemma B.3. (Gerschgorin Disc) (Bhatia, 2013)** *Let* $\mathbf{A} \in \mathbb{C}^{n \times n}$*, with entries* $a_{ij}$*. For any eigenvalue* $\lambda$*, there exits* $i$ *and the corresponding Gerschgorin disc* $D(a_{ii}, R_i) \subseteq \mathbb{C}$ *such that* $\lambda$ *lies in this disc, i.e.*

$$|\lambda - a_{ii}| \leq \sum_{j \neq i}^{n} |a_{ij}|.$$

Lemma B.3 describes the estimated range of eigenvalues. Now we start to derive the Neumann Series expansion of the solution of GSD as follows.

**Lemma B.4.** *Let* $\mathbf{A} \in \{0,1\}^{n \times n}$ *be the adjacency matrix of a graph and* $\widetilde{\mathcal{A}} = \widetilde{\mathbf{D}}^{-\frac{1}{2}} \widetilde{\mathbf{A}} \widetilde{\mathbf{D}}^{-\frac{1}{2}}$ *or* $\widetilde{\mathcal{A}} = \widetilde{\mathbf{D}}^{-1} \widetilde{\mathbf{A}}$*, then*

$$(\mathbf{I} - \frac{\lambda}{\lambda + 1} \widetilde{\mathcal{A}})^{-1} = \sum_{k=0}^{\infty} \left( \frac{\lambda}{\lambda + 1} \widetilde{\mathcal{A}} \right)^k.$$

*Proof.* We first prove that $\rho(\widetilde{\mathcal{A}}) \leq 1$ where $\widetilde{\mathcal{A}} = \widetilde{\mathbf{D}}^{-\frac{1}{2}} \widetilde{\mathbf{A}} \widetilde{\mathbf{D}}^{-\frac{1}{2}}$. Let $\lambda$ be the eigenvalue of $\widetilde{\mathcal{A}}$, and $\mathbf{v}$ be the corresponding eigenvector. Then we have

$$\left( \widetilde{\mathbf{D}}^{-\frac{1}{2}} \widetilde{\mathbf{A}} \widetilde{\mathbf{D}}^{-\frac{1}{2}} \right) \mathbf{v} = \lambda \mathbf{v} \Longrightarrow \widetilde{\mathbf{D}}^{-\frac{1}{2}} \left( \widetilde{\mathbf{D}}^{-\frac{1}{2}} \widetilde{\mathbf{A}} \widetilde{\mathbf{D}}^{-\frac{1}{2}} \right) \mathbf{v} = \lambda \widetilde{\mathbf{D}}^{-\frac{1}{2}} \mathbf{v}$$

$$\Longrightarrow \left( \widetilde{\mathbf{D}}^{-1} \widetilde{\mathbf{A}} \right) \widetilde{\mathbf{D}}^{-\frac{1}{2}} \mathbf{v} = \lambda \widetilde{\mathbf{D}}^{-\frac{1}{2}} \mathbf{v},$$

which means $(\lambda, \widetilde{\mathbf{D}}^{-\frac{1}{2}} \mathbf{v})$ is the eigen-pair of $\widetilde{\mathbf{D}}^{-1} \mathbf{A}$. By Lemma B.3, there exists $i$, such that

$$\left| \lambda - \left( \widetilde{\mathbf{D}}^{-1} \widetilde{\mathbf{A}} \right)_{ii} \right| \leq \sum_{j \neq i} \left| \left( \widetilde{\mathbf{D}}^{-1} \widetilde{\mathbf{A}} \right)_{ij} \right|$$

$$\Longrightarrow \left( \widetilde{\mathbf{D}}^{-1} \widetilde{\mathbf{A}} \right)_{ii} - \sum_{j \neq i} \left| \left( \widetilde{\mathbf{D}}^{-1} \widetilde{\mathbf{A}} \right)_{ij} \right| \leq \lambda \leq \left( \widetilde{\mathbf{D}}^{-1} \widetilde{\mathbf{A}} \right)_{ii} + \sum_{j \neq i} \left| \left( \widetilde{\mathbf{D}}^{-1} \widetilde{\mathbf{A}} \right)_{ij} \right|.$$

Since $\left( \widetilde{\mathbf{D}}^{-1} \widetilde{\mathbf{A}} \right)_{ij} > 0$ and $\sum_j \left| \left( \widetilde{\mathbf{D}}^{-1} \widetilde{\mathbf{A}} \right)_{ij} \right| = \sum_j \left( \widetilde{\mathbf{D}}^{-1} \widetilde{\mathbf{A}} \right)_{ij} = 1$, obviously

$$-1 < \left( \widetilde{\mathbf{D}}^{-1} \widetilde{\mathbf{A}} \right)_{ii} - \sum_{j \neq i} \left| \left( \widetilde{\mathbf{D}}^{-1} \widetilde{\mathbf{A}} \right)_{ij} \right| \leq \lambda \leq \left( \widetilde{\mathbf{D}}^{-1} \widetilde{\mathbf{A}} \right)_{ii} + \sum_{j \neq i} \left| \left( \widetilde{\mathbf{D}}^{-1} \widetilde{\mathbf{A}} \right)_{ij} \right| = 1.$$

So if $\widetilde{\mathcal{A}} = \widetilde{\mathbf{D}}^{-\frac{1}{2}} \widetilde{\mathbf{A}} \widetilde{\mathbf{D}}^{-\frac{1}{2}}$, we have $\rho(\widetilde{\mathcal{A}}) \leq 1$. When $\widetilde{\mathcal{A}} = \widetilde{\mathbf{D}}^{-1} \widetilde{\mathbf{A}}$, we denote $(\lambda, \mathbf{v})$ as the eigen-pair of $\widetilde{\mathbf{D}}^{-1} \mathbf{A}$. Similarly, by Lemma B.3, there exists $i$, such that

$$\left| \lambda - \left( \widetilde{\mathbf{D}}^{-1} \widetilde{\mathbf{A}} \right)_{ii} \right| \leq \sum_{j \neq i} \left| \left( \widetilde{\mathbf{D}}^{-1} \widetilde{\mathbf{A}} \right)_{ij} \right|$$

$$\Longrightarrow \left( \widetilde{\mathbf{D}}^{-1} \widetilde{\mathbf{A}} \right)_{ii} - \sum_{j \neq i} \left| \left( \widetilde{\mathbf{D}}^{-1} \widetilde{\mathbf{A}} \right)_{ij} \right| \leq \lambda \leq \left( \widetilde{\mathbf{D}}^{-1} \widetilde{\mathbf{A}} \right)_{ii} + \sum_{j \neq i} \left| \left( \widetilde{\mathbf{D}}^{-1} \widetilde{\mathbf{A}} \right)_{ij} \right|.$$

Obviously, we can get the same conclusion for $\widetilde{\mathcal{A}} = \widetilde{\mathbf{D}}^{-1} \widetilde{\mathbf{A}}$. So it is true for $\rho \left( \frac{\lambda}{\lambda+1} \widetilde{\mathcal{A}} \right) \leq \frac{\lambda}{\lambda+1} < 1$

By Lemma B.2, we get the result $(\mathbf{I} - \frac{\lambda}{\lambda+1} \widetilde{\mathcal{A}})^{-1} = \sum_{k=0}^{\infty} \left( \frac{\lambda}{\lambda+1} \widetilde{\mathcal{A}} \right)^k$, which finishes the proof. $\quad\square$

By Lemma B.4, we approximate the inverse matrix $(\mathbf{I} + \lambda \tilde{\mathbf{L}})^{-1}$ up to $S$-th order with

$$\left( \mathbf{I} + \lambda \tilde{\mathbf{L}} \right)^{-1} = \frac{1}{\lambda + 1} \left( \mathbf{I} - \frac{\lambda}{\lambda + 1} \widetilde{\mathcal{A}} \right)^{-1} \approx \frac{1}{\lambda + 1} \sum_{s=0}^{S} \left( \frac{\lambda}{\lambda + 1} \widetilde{\mathcal{A}} \right)^s.$$

## C  THE ROW SUMMATION OF THE NEUMANN SERIES

We provide the derivations of the row sum of $\widetilde{\boldsymbol{\mathcal{A}}}_S$ in this section. Before we derive the row summation of $\widetilde{\boldsymbol{\mathcal{A}}}_S$, we first derive the row summation of $\widetilde{\boldsymbol{\mathcal{A}}}^k$.

**Lemma C.1.** *Consider a probability matrix* $\mathbf{P} \in \mathbb{R}^{n \times n}$, *where* $\mathbf{P}_{ij} \geq 0$. *Besides, for all* $i$, *we have* $\sum_{j=1}^n \mathbf{P}_{ij} = 1$. *Then for any* $s \in \mathbb{Z}_+$, *we have* $\sum_{j=1}^n \mathbf{P}_{ij}^s = 1$,

*Proof.* We give a proof by induction on $k$.
**Base case:** When $k = 1$, the case is true.
**Inductive step:** Assume the induction hypothesis that for a particular $k$, the single case n = k holds, meaning $\mathbf{P}^k$ is true:

$$\forall i, \sum_{j=1}^n \mathbf{P}_{ij}^k = 1.$$

As $\mathbf{P}^{k+1} = \mathbf{P}^k \mathbf{P}$, so we have

$$\sum_{j=1}^n \mathbf{P}_{ij}^{k+1} = \sum_{j=1}^n \sum_{k=1}^n \mathbf{P}_{ik}^k \mathbf{P}_{kj} = \sum_{k=1}^n \sum_{j=1}^n \mathbf{P}_{ik}^k \mathbf{P}_{kj} = \sum_{k=1}^n \mathbf{P}_{ik}^k \left( \sum_{j=1}^n \mathbf{P}_{kj} \right) = \sum_{k=1}^n \mathbf{P}_{ik}^k = 1,$$

which finishes the proof. $\square$

Lemma C.1 describes the row summation of $\widetilde{\boldsymbol{\mathcal{A}}}^k$ is 1. Now we can obtain the row summation for $\widetilde{\boldsymbol{\mathcal{A}}}_S$. Then for any $i$, we have

$$
\begin{aligned}
\sum_{j=1}^n \left[ \widetilde{\boldsymbol{\mathcal{A}}}_S \right]_{ij} &= \frac{1}{\lambda+1} \sum_{s=0}^S \left( \frac{\lambda}{\lambda+1} \left[ \widetilde{\boldsymbol{\mathcal{A}}} \right]_{ij} \right)^s \\
&= \frac{1}{\lambda+1} \sum_{s=0}^S \left( \frac{\lambda}{\lambda+1} \right)^s \\
&= 1 - \left( \frac{\lambda}{\lambda+1} \right)^{S+1}.
\end{aligned}
\tag{19}
$$

## D  PROOF OF LEMMA 1

We provide the details of proof of Lemma 1. We first introduce the General Hoeffding Inequality (Hoeffding, 1994), which is essential for bounding $\left\| \widetilde{\boldsymbol{\mathcal{A}}}_S \boldsymbol{\eta} \right\|_F^2$.

**Lemma D.1. (General Hoeffding Inequality (Hoeffding, 1994))** *Suppose that the variables* $X_1, \cdots, X_n$ *are independent, and* $X_i$ *has mean* $\mu_i$ *and sub-Gaussian parameter* $\sigma_i$. *Then for all* $t \geq 0$, *we have*

$$\mathbb{P} \left[ \sum_{i=1}^n (X_i - \mu_i) \geq t \right] \leq \exp \left\{ -\frac{t^2}{2 \sum_{i=1}^n \sigma_i^2} \right\}.
\tag{20}$$

Now let's prove Lemma 1.

*Proof of Lemma 1.* For any entry $\left[ \widetilde{\boldsymbol{\mathcal{A}}}_S \boldsymbol{\eta} \right]_{ij} = \sum_{p=1}^n \left( \widetilde{\boldsymbol{\mathcal{A}}}_S \right)_{ip} \boldsymbol{\eta}_{pj}$, where $\boldsymbol{\eta}_{pj}$ is a sub-Gaussian variable with parameter $\sigma^2$. By the General Hoeffding inequality D.1, we have

$$\mathbb{P} \left( \left| \left[ \frac{1}{\lambda+1} \sum_{s=0}^S \left( \frac{\lambda}{\lambda+1} \widetilde{\boldsymbol{\mathcal{A}}}_S \right)^s \boldsymbol{\eta} \right]_{ij} \right| \geq t \right) \leq 2 \exp \left\{ -\frac{n t^2}{2 \tau \left( 1 - \left( \frac{\lambda}{\lambda+1} \right)^{S+1} \right)^2 \sigma^2} \right\}.
\tag{21}$$

where $\tau = \max_i \tau_i$ and $\tau_i = n \sum_{j=1}^n \left[ \widetilde{\boldsymbol{\mathcal{A}}}_S \right]_{ij}^2 \Big/ \left( 1 - \left( \frac{\lambda}{\lambda+1} \right)^{S+1} \right)^2$.

Applying union bound (Vershynin, 2010) to all possible pairs of $i \in [n]$, $j \in [n]$, we get

$$\mathbb{P}\left( \left\| \widetilde{\boldsymbol{\mathcal{A}}}_S \boldsymbol{\eta} \right\|_{\infty,\infty} \geq t \right) \leq \sum_{i,j} \mathbb{P}\left( \left[ \widetilde{\boldsymbol{\mathcal{A}}}_S \boldsymbol{\eta} \right]_{ij} \geq t \right) \leq 2n^2 \exp\left\{ -\frac{nt^2}{2\tau \left( 1 - \left( \frac{\lambda}{\lambda+1} \right)^{S+1} \right)^2 \sigma^2} \right\}. \tag{22}$$

Applying union bound again, we have

$$\mathbb{P}\left( \left\| \widetilde{\boldsymbol{\mathcal{A}}}_S \boldsymbol{\eta} \right\|_F^2 \geq t \right) \leq \sum_{i,j} \mathbb{P}\left( \left\| \widetilde{\boldsymbol{\mathcal{A}}}_S \boldsymbol{\eta} \right\|_{\infty,\infty} \geq \sqrt{t} \right) \leq 2n^4 \exp\left\{ -\frac{nt}{2\tau \left( 1 - \left( \frac{\lambda}{\lambda+1} \right)^{S+1} \right)^2 \sigma^2} \right\}. \tag{23}$$

Choose $t = 2\tau \left( 1 - \left( \frac{\lambda}{\lambda+1} \right)^{S+1} \right)^2 (4\log n + \log 2d)/n$ and with probability $1 - 1/d$, we have

$$\left\| \widetilde{\boldsymbol{\mathcal{A}}}_S \boldsymbol{\eta} \right\|_F^2 \leq \frac{2\tau \left( 1 - \left( \frac{\lambda}{\lambda+1} \right)^{S+1} \right)^2 \sigma^2 (4\log n + \log 2d)}{n}, \tag{24}$$

which finishes the proof. $\qquad\square$

## E   PROOF OF THE MAIN THEOREM 1

We provide the details of proof of main theorem 1.

**[Restatement of Theorem 1]** *Under Assumptions 1,2,3,4, let $\mathbf{W}_f^{(k)}$ denote the $k$-th step gradient descent solution for $\min_{\mathbf{W}} f(\mathbf{W})$ with step size $\alpha \leq 1/L$, with probability $1 - 1/d$ we have*

$$g\left( \mathbf{W}_f^{(k)} \right) - g\left( \mathbf{W}_g^* \right) \leq \mathcal{O}\left( \frac{1}{2k\alpha} \right) + \mathcal{O}\left( \frac{\tau \log n}{n} \right), \tag{25}$$

*where $\mathbf{W}_g^* = \arg\min_{\mathbf{W}} g(\mathbf{W})$ is the optimal solution of the clean loss function $g(\mathbf{W})$, $\tau$ is the high-order graph connectivity factor, and $n$ is the number of nodes of a graph.*

*Proof.* By the definition of $L$-smooth, we can obtain the following inequality:

$$f(\mathbf{W}_f') \leq f(\mathbf{W}_f) + \langle \nabla f(\mathbf{W}_f), \mathbf{W}_f' - \mathbf{W}_f \rangle + \frac{1}{2} L \|\mathbf{W}_f' - \mathbf{W}_f\|_F^2. \tag{26}$$

Let's use the gradient descent algorithm with $\mathbf{W}_f' = \mathbf{W}_f^+ = \mathbf{W}_f - \alpha \nabla f(\mathbf{W}_f)$. We then get:

$$\begin{aligned}
f\left( \mathbf{W}_f^+ \right) &\leq f(\mathbf{W}_f) + \langle \nabla f(\mathbf{W}_f), \mathbf{W}_f^+ - \mathbf{W}_f \rangle + \frac{1}{2} L \left\| \mathbf{W}_f^+ - \mathbf{W}_f \right\|_F^2 \\
&= f(\mathbf{W}_f) + \langle \nabla f(\mathbf{W}_f), \mathbf{W}_f - \alpha \nabla f(\mathbf{W}_f) - \mathbf{W}_f \rangle + \frac{1}{2} L \|\mathbf{W}_f - \alpha \nabla f(\mathbf{W}_f) - \mathbf{W}_f\|_F^2 \\
&= f(\mathbf{W}_f) - \langle \nabla f(\mathbf{W}_f), \alpha \nabla f(\mathbf{W}_f) \rangle + \frac{1}{2} L \|\alpha \nabla f(\mathbf{W}_f)\|_F^2 \\
&= f(\mathbf{W}_f) - \alpha \|\nabla f(\mathbf{W}_f)\|_F^2 + \frac{1}{2} L \alpha^2 \|\nabla f(\mathbf{W}_f)\|_F^2 \\
&= f(\mathbf{W}_f) - \left( 1 - \frac{1}{2} L\alpha \right) \alpha \|\nabla f(\mathbf{W}_f)\|_F^2.
\end{aligned} \tag{27}$$

With the fixed step size $\alpha \leq 1/L$, we know that $-(1 - \frac{1}{2}L\alpha) = \frac{1}{2}L\alpha - 1 \leq \frac{1}{2}L(1/L) - 1 = \frac{1}{2} - 1 = -\frac{1}{2}$. Plugging this into Eq. (27), we have the following inequality:

$$f\left(\mathbf{W}_f^+\right) \leq f(\mathbf{W}_f) - \frac{1}{2}\alpha\|\nabla f(\mathbf{W}_f)\|_F^2. \tag{28}$$

If we choose $t$ to be small enough such that $t \leq 1/L$, this inequality implies that the loss function value strictly decreases under each iteration of gradient descent since $\|\nabla f(\mathbf{W}_f)\|$ is positive unless $\nabla f(\mathbf{W}_f) = 0$ e.g. $\mathbf{W}_f = \mathbf{W}_f^*$, where $\mathbf{W}_f$ reaches $\mathbf{W}_f^*$.

Now, let's bound the loss function value $f(\mathbf{W}_f^+)$. Since $f$ is convex, we can write

$$f(\mathbf{W}_f) \leq f\left(\mathbf{W}_g^*\right) + \langle\nabla f(\mathbf{W}_f), \mathbf{W}_f - \mathbf{W}_g^*\rangle. \tag{29}$$

Introducing this inequality into Eq. (28), we can obtain the following:

$$\begin{aligned}
f\left(\mathbf{W}_f^+\right) - f\left(\mathbf{W}_g^*\right) &\leq \langle\nabla f(\mathbf{W}_f), \mathbf{W}_f - \mathbf{W}_g^*\rangle - \frac{\alpha}{2}\|\nabla f(\mathbf{W}_f)\|_F^2 \\
&\leq \frac{1}{2\alpha}\left(2\alpha\langle\nabla f(\mathbf{W}_f), \mathbf{W}_f - \mathbf{W}_g^*\rangle - \alpha^2\|\nabla f(\mathbf{W}_f)\|_F^2\right) \\
&\leq \frac{1}{2\alpha}\left(2\alpha\langle\nabla f(\mathbf{W}_f), \mathbf{W}_f - \mathbf{W}_g^*\rangle - \alpha^2\|\nabla f(\mathbf{W}_f)\|_F^2 - \left\|\mathbf{W}_f - \mathbf{W}_g^*\right\|_F^2\right) \\
&\quad + \frac{1}{2\alpha}\left\|\mathbf{W}_f - \mathbf{W}_g^*\right\|_F^2 \\
&\leq \frac{1}{2\alpha}\left(\left\|\mathbf{W}_f - \mathbf{W}_g^*\right\|_F^2 - \left\|\mathbf{W}_f - \alpha\nabla f(\mathbf{W}_f) - \mathbf{W}_g^*\right\|_F^2\right).
\end{aligned} \tag{30}$$

Notice that by the definition of gradient descent update, we have $\mathbf{W}_f^+ = \mathbf{W}_f - \alpha\nabla f(\mathbf{W}_f)$. Plugging this into the final inequality of Eq. (30), we can get:

$$f\left(\mathbf{W}_f^+\right) - f\left(\mathbf{W}_g^*\right) \leq \frac{1}{2\alpha}\left(\left\|\mathbf{W}_f - \mathbf{W}_g^*\right\|_F^2 - \left\|\mathbf{W}_f^+ - \mathbf{W}_g^*\right\|_F^2\right). \tag{31}$$

This inequality holds for $\mathbf{W}_f^+$ on every iteration of gradient descent. Summing over iterations, we get:

$$\begin{aligned}
\sum_{i=1}^k\left[f\left(\mathbf{W}_f^{(i)}\right) - f\left(\mathbf{W}_g^*\right)\right] &\leq \sum_{i=1}^k \frac{1}{2\alpha}\left(\left\|\mathbf{W}_f^{(i-1)} - \mathbf{W}_g^*\right\|_F^2 - \left\|\mathbf{W}_f^{(i)} - \mathbf{W}_g^*\right\|_F^2\right) \\
&= \frac{1}{2\alpha}\left(\left\|\mathbf{W}_f^{(0)} - \mathbf{W}_g^*\right\|_F^2 - \left\|\mathbf{W}_f^{(k)} - \mathbf{W}_g^*\right\|_F^2\right) \\
&\leq \frac{1}{2\alpha}\left(\left\|\mathbf{W}_f^{(0)} - \mathbf{W}_g^*\right\|_F^2\right).
\end{aligned} \tag{32}$$

With the inequality of Eq. (29), we know that $f(\mathbf{W}_f)$ strictly decreases over each iteration. So we have following:

$$\begin{aligned}
f\left(\mathbf{W}_f^{(k)}\right) - f\left(\mathbf{W}_g^*\right) &\leq \frac{1}{k}\left[\sum_{i=1}^k f\left(\mathbf{W}_f^{(i)}\right) - f\left(\mathbf{W}_g^*\right)\right] \\
&\leq \frac{1}{2k\alpha}\left(\left\|\mathbf{W}_f^{(0)} - \mathbf{W}_g^*\right\|_F^2\right)
\end{aligned} \tag{33}$$

Equivalently, we have the inequality for the loss function $g(\mathbf{W}_f)$:

$$
\begin{aligned}
g\left(\mathbf{W}_f^{(k)}\right) - g\left(\mathbf{W}_g^*\right) = {}& f\left(\mathbf{W}_f^{(k)}\right) - f\left(\mathbf{W}_g^*\right) \\
& + 2\langle \widetilde{\boldsymbol{\mathcal{A}}}_S \boldsymbol{\eta} \mathbf{W}_g^*, \widetilde{\boldsymbol{\mathcal{A}}}_S \mathbf{X}^* \mathbf{W}_g^* - \mathbf{Y}\rangle + \langle \widetilde{\boldsymbol{\mathcal{A}}}_S \boldsymbol{\eta} \mathbf{W}_g^*, \widetilde{\boldsymbol{\mathcal{A}}}_S \boldsymbol{\eta} \mathbf{W}_g^*\rangle \\
& - 2\langle \widetilde{\boldsymbol{\mathcal{A}}}_S \boldsymbol{\eta} \mathbf{W}_f^{(k)}, \widetilde{\boldsymbol{\mathcal{A}}}_S \mathbf{X}^* \mathbf{W}_f^{(k)} - \mathbf{Y}\rangle - \langle \widetilde{\boldsymbol{\mathcal{A}}}_S \boldsymbol{\eta} \mathbf{W}_f^{(k)}, \widetilde{\boldsymbol{\mathcal{A}}}_S \boldsymbol{\eta} \mathbf{W}_f^{(k)}\rangle \\
\leq {}& \frac{1}{2k\alpha}\left(\left\|\mathbf{W}_f^{(0)} - \mathbf{W}_g^*\right\|_F^2\right) \\
& + \left\|\widetilde{\boldsymbol{\mathcal{A}}}_S \boldsymbol{\eta}\right\|_F^2 \left\|\mathbf{W}_g^*\right\|_F^2 \left(2\left\|\widetilde{\boldsymbol{\mathcal{A}}}_S \mathbf{X}^* \mathbf{W}_g^* - \mathbf{Y}\right\|_F^2 + \left\|\widetilde{\boldsymbol{\mathcal{A}}}_S \boldsymbol{\eta}\right\|_F^2 \left\|\mathbf{W}_g^*\right\|_F^2\right) \\
& + \left\|\widetilde{\boldsymbol{\mathcal{A}}}_S \boldsymbol{\eta}\right\|_F^2 \left\|\mathbf{W}_f^{(k)}\right\|_F^2 \left(2\left\|\widetilde{\boldsymbol{\mathcal{A}}}_S \mathbf{X} \mathbf{W}_f^{(k)} - \mathbf{Y}\right\|_F^2 + \left\|\widetilde{\boldsymbol{\mathcal{A}}}_S \boldsymbol{\eta}\right\|_F^2 \left\|\mathbf{W}_f^{(k)}\right\|_F^2\right) \\
\leq {}& \mathcal{O}\left(\frac{1}{2k\alpha}\right) + \mathcal{O}\left(\frac{\tau \log n}{n}\right),
\end{aligned}
\tag{34}
$$

which finishes the proof. $\qquad\square$

## F  MORE DETAILS ON EQUATION (1).

We provide more details on how to obtain Equation (1).

Note that if we set $\widetilde{\mathbf{L}} = \mathbf{I} - \widetilde{\mathbf{D}}^{-\frac{1}{2}} \widetilde{\mathbf{A}} \widetilde{\mathbf{D}}^{-\frac{1}{2}}$, we have $\text{tr}\left(\mathbf{F}^\top \widetilde{\mathbf{L}} \mathbf{F}\right) = \text{tr}\left(\mathbf{F}^\top (\mathbf{I} - \widetilde{\mathbf{D}}^{-\frac{1}{2}} \widetilde{\mathbf{A}} \widetilde{\mathbf{D}}^{-\frac{1}{2}})\mathbf{F}\right) = \text{tr}\left(\mathbf{F}^\top \mathbf{F}\right) - \text{tr}\left(\mathbf{F}^\top \widetilde{\mathbf{D}}^{-\frac{1}{2}} \widetilde{\mathbf{A}} \widetilde{\mathbf{D}}^{-\frac{1}{2}} \mathbf{F}\right) = \text{tr}\left(\mathbf{F}\mathbf{F}^\top\right) - \text{tr}\left(\widetilde{\mathbf{D}}^{-\frac{1}{2}} \widetilde{\mathbf{A}} \widetilde{\mathbf{D}}^{-\frac{1}{2}} \mathbf{F}\mathbf{F}^\top\right)$. On the other hand, if we set $\widetilde{\mathbf{L}} = \mathbf{I} - \widetilde{\mathbf{D}}^{-1}\widetilde{\mathbf{A}}$, we have $\text{tr}\left(\mathbf{F}^\top \widetilde{\mathbf{L}} \mathbf{F}\right) = \text{tr}\left(\mathbf{F}^\top (\mathbf{I} - \widetilde{\mathbf{D}}^{-1}\widetilde{\mathbf{A}})\mathbf{F}\right) = \text{tr}\left(\mathbf{F}^\top \mathbf{F}\right) - \text{tr}\left(\mathbf{F}^\top \widetilde{\mathbf{D}}^{-1}\widetilde{\mathbf{A}}\mathbf{F}\right) = \text{tr}\left(\mathbf{F}\mathbf{F}^\top\right) - \text{tr}\left(\widetilde{\mathbf{D}}^{-1}\widetilde{\mathbf{A}}\mathbf{F}\mathbf{F}^\top\right)$. We denote $\mathbf{F} = \begin{bmatrix} \mathbf{F}_1 \\ \vdots \\ \mathbf{F}_n \end{bmatrix}$ and $\mathbf{F}^\top = \begin{bmatrix} \mathbf{F}_1^\top \cdots \mathbf{F}_n^\top \end{bmatrix}$, where $\mathbf{F}_i = [\mathbf{F}_{i1} \cdots \mathbf{F}_{id}]$, then we have $\text{tr}\left(\mathbf{F}\mathbf{F}^\top\right) = \sum_{i=1}^n \mathbf{F}_i \mathbf{F}_i^\top$.

When $\widetilde{\mathbf{L}} = \mathbf{I} - \widetilde{\mathbf{D}}^{-\frac{1}{2}} \widetilde{\mathbf{A}} \widetilde{\mathbf{D}}^{-\frac{1}{2}}$, we have

$$
\begin{aligned}
& \text{tr}\left(\widetilde{\mathbf{D}}^{-\frac{1}{2}} \widetilde{\mathbf{A}} \widetilde{\mathbf{D}}^{-\frac{1}{2}} \mathbf{F}\mathbf{F}^\top\right) \\
= {}& \text{tr}\left(
\begin{bmatrix}
\frac{\mathbf{A}_{11}}{\sqrt{d_1+1}\sqrt{d_1+1}} & \frac{\mathbf{A}_{12}}{\sqrt{d_1+1}\sqrt{d_2+1}} & \cdots & \frac{\mathbf{A}_{1n}}{\sqrt{d_1+1}\sqrt{d_n+1}} \\
\frac{\mathbf{A}_{21}}{\sqrt{d_2+1}\sqrt{d_1+1}} & \frac{\mathbf{A}_{22}}{\sqrt{d_2+1}\sqrt{d_2+1}} & \cdots & \frac{\mathbf{A}_{2n}}{\sqrt{d_2+1}\sqrt{d_n+1}} \\
\vdots & \ddots & \ddots & \vdots \\
\frac{\mathbf{A}_{n1}}{\sqrt{d_n+1}\sqrt{d_1+1}} & \frac{\mathbf{A}_{n2}}{\sqrt{d_n+1}\sqrt{d_2+1}} & \cdots & \frac{\mathbf{A}_{nn}}{\sqrt{d_n+1}\sqrt{d_n+1}}
\end{bmatrix}
\begin{bmatrix}
\mathbf{F}_1\mathbf{F}_1^\top & \mathbf{F}_1\mathbf{F}_2^\top & \cdots & \mathbf{F}_1\mathbf{F}_n^\top \\
\mathbf{F}_2\mathbf{F}_1^\top & \mathbf{F}_2\mathbf{F}_2^\top & \cdots & \mathbf{F}_2\mathbf{F}_n^\top \\
\vdots & \ddots & \ddots & \vdots \\
\mathbf{F}_n\mathbf{F}_1^\top & \mathbf{F}_n\mathbf{F}_2^\top & \cdots & \mathbf{F}_n\mathbf{F}_n^\top
\end{bmatrix}
\right) \\
= {}& \sum_{i=1}^n \sum_{j=1}^n \frac{\mathbf{A}_{ij}}{\sqrt{d_i+1}\sqrt{d_j+1}} \mathbf{F}_j \mathbf{F}_i^\top.
\end{aligned}
$$

On the other hand, when $\widetilde{\mathbf{L}} = \mathbf{I} - \widetilde{\mathbf{D}}^{-1}\widetilde{\mathbf{A}}$, we have

$$
\begin{aligned}
& \text{tr}\left(\widetilde{\mathbf{D}}^{-1}\widetilde{\mathbf{A}}\mathbf{F}\mathbf{F}^\top\right) \\
= {}& \text{tr}\left(
\begin{bmatrix}
\frac{\mathbf{A}_{11}}{d_1+1} & \frac{\mathbf{A}_{12}}{d_1+1} & \cdots & \frac{\mathbf{A}_{1n}}{d_1+1} \\
\frac{\mathbf{A}_{21}}{d_2+1} & \frac{\mathbf{A}_{22}}{d_2+1} & \cdots & \frac{\mathbf{A}_{2n}}{d_2+1} \\
\vdots & \ddots & \ddots & \vdots \\
\frac{\mathbf{A}_{n1}}{d_n+1} & \frac{\mathbf{A}_{n2}}{d_n+1} & \cdots & \frac{\mathbf{A}_{nn}}{d_n+1}
\end{bmatrix}
\begin{bmatrix}
\mathbf{F}_1\mathbf{F}_1^\top & \mathbf{F}_1\mathbf{F}_2^\top & \cdots & \mathbf{F}_1\mathbf{F}_n^\top \\
\mathbf{F}_2\mathbf{F}_1^\top & \mathbf{F}_2\mathbf{F}_2^\top & \cdots & \mathbf{F}_2\mathbf{F}_n^\top \\
\vdots & \ddots & \ddots & \vdots \\
\mathbf{F}_n\mathbf{F}_1^\top & \mathbf{F}_n\mathbf{F}_2^\top & \cdots & \mathbf{F}_n\mathbf{F}_n^\top
\end{bmatrix}
\right) \\
= {}& \sum_{i=1}^n \sum_{j=1}^n \frac{\mathbf{A}_{ij}}{d_i+1} \mathbf{F}_j \mathbf{F}_i^\top.
\end{aligned}
$$

So when $\widetilde{\mathbf{L}} = \mathbf{I} - \widetilde{\mathbf{D}}^{-\frac{1}{2}}\widetilde{\mathbf{A}}\widetilde{\mathbf{D}}^{-\frac{1}{2}}$, we have

$$\mathrm{tr}\left(\mathbf{F}^\top\widetilde{\mathbf{L}}\mathbf{F}\right) \quad \left(\widetilde{\mathbf{L}} = \mathbf{I} - \widetilde{\mathbf{D}}^{-\frac{1}{2}}\widetilde{\mathbf{A}}\widetilde{\mathbf{D}}^{-\frac{1}{2}}\right)$$

$$= \mathrm{tr}\left(\mathbf{F}^\top(\mathbf{I} - \widetilde{\mathbf{D}}^{-\frac{1}{2}}\widetilde{\mathbf{A}}\widetilde{\mathbf{D}}^{-\frac{1}{2}})\mathbf{F}\right)$$

$$= \mathrm{tr}\left(\mathbf{F}\mathbf{F}^\top\right) - \mathrm{tr}\left(\widetilde{\mathbf{D}}^{-\frac{1}{2}}\widetilde{\mathbf{A}}\widetilde{\mathbf{D}}^{-\frac{1}{2}}\mathbf{F}\mathbf{F}^\top\right)$$

$$= \sum_{i=1}^n \mathbf{F}_i\mathbf{F}_i^\top - \sum_{i=1}^n\sum_{j=1}^n \frac{\mathbf{A}_{ij}}{\sqrt{d_i+1}\sqrt{d_j+1}}\mathbf{F}_j\mathbf{F}_i^\top$$

$$= \frac{1}{2}\sum_{i=1}^n \mathbf{F}_i\mathbf{F}_i^\top + \frac{1}{2}\sum_{j=1}^n \mathbf{F}_j\mathbf{F}_j^\top - \sum_{i=1}^n\sum_{j=1}^n \frac{\mathbf{A}_{ij}}{\sqrt{d_i+1}\sqrt{d_j+1}}\mathbf{F}_j\mathbf{F}_i^\top$$

$$= \frac{1}{2}\left(\sum_{i=1}^n \mathbf{F}_i\mathbf{F}_i^\top + \sum_{j=1}^n \mathbf{F}_j\mathbf{F}_j^\top - 2\sum_{i=1}^n\sum_{j=1}^n \frac{\mathbf{A}_{ij}}{\sqrt{d_i+1}\sqrt{d_j+1}}\mathbf{F}_j\mathbf{F}_i^\top\right)$$

$$= \frac{1}{2}\left(\sum_{i=1}^n\sum_{j=1}^n \frac{\mathbf{A}_{ij}\mathbf{F}_i\mathbf{F}_i^\top}{d_i+1} + \sum_{i=1}^n\sum_{j=1}^n \frac{\mathbf{A}_{ij}\mathbf{F}_j\mathbf{F}_j^\top}{d_j+1} - 2\sum_{i=1}^n\sum_{j=1}^n \frac{\mathbf{A}_{ij}}{\sqrt{d_i+1}\sqrt{d_j+1}}\mathbf{F}_j\mathbf{F}_i^\top\right) \text{ undirected graph}$$

$$= \frac{1}{2}\left(\sum_{i=1}^n\sum_{j=1}^n \left(\frac{\mathbf{A}_{ij}\mathbf{F}_i\mathbf{F}_i^\top}{d_i+1} + \frac{\mathbf{A}_{ij}\mathbf{F}_j\mathbf{F}_j^\top}{d_j+1} - \frac{\mathbf{A}_{ij}}{\sqrt{d_i+1}\sqrt{d_j+1}}\mathbf{F}_j\mathbf{F}_i^\top - \frac{\mathbf{A}_{ij}}{\sqrt{d_i+1}\sqrt{d_j+1}}\mathbf{F}_i\mathbf{F}_j^\top\right)\right)$$

$$= \frac{1}{2}\left(\sum_{i=1}^n\sum_{j=1}^n \mathbf{A}_{ij}\left(\frac{\mathbf{F}_i\mathbf{F}_i^\top}{d_i+1} + \frac{\mathbf{F}_j\mathbf{F}_j^\top}{d_j+1} - \frac{\mathbf{F}_j\mathbf{F}_i^\top}{\sqrt{d_i+1}\sqrt{d_j+1}} - \frac{\mathbf{F}_i\mathbf{F}_j^\top}{\sqrt{d_i+1}\sqrt{d_j+1}}\right)\right)$$

$$= \frac{1}{2}\left(\sum_{i=1}^n\sum_{j=1}^n \mathbf{A}_{ij}\left(\frac{\mathbf{F}_i}{\sqrt{d_i+1}} - \frac{\mathbf{F}_j}{\sqrt{d_j+1}}\right)\left(\frac{\mathbf{F}_i^\top}{\sqrt{d_i+1}} - \frac{\mathbf{F}_j^\top}{\sqrt{d_j+1}}\right)\right)$$

$$= \frac{1}{2}\left(\sum_{i=1}^n\sum_{j=1}^n \mathbf{A}_{ij}\left\|\frac{\mathbf{F}_i}{\sqrt{d_i+1}} - \frac{\mathbf{F}_j}{\sqrt{d_j+1}}\right\|_2^2\right) = \sum_{(i,j)\in\mathcal{E}} \mathbf{A}_{ij}\left\|\frac{\mathbf{F}_i}{\sqrt{d_i+1}} - \frac{\mathbf{F}_j}{\sqrt{d_j+1}}\right\|_2^2.$$

On the other hand, when $\widetilde{\mathbf{L}} = \mathbf{I} - \widetilde{\mathbf{D}}^{-1}\widetilde{\mathbf{A}}$, we have

$$\mathrm{tr}\left(\mathbf{F}^\top\widetilde{\mathbf{L}}\mathbf{F}\right) \quad \left(\widetilde{\mathbf{L}} = \mathbf{I} - \widetilde{\mathbf{D}}^{-1}\widetilde{\mathbf{A}}\right)$$

$$= \mathrm{tr}\left(\mathbf{F}^\top(\mathbf{I} - \widetilde{\mathbf{D}}^{-1}\widetilde{\mathbf{A}})\mathbf{F}\right)$$

$$= \sum_{i=1}^n \mathbf{F}_i\mathbf{F}_i^\top - \sum_{i=1}^n\sum_{j=1}^n \frac{\mathbf{A}_{ij}}{d_i+1}\mathbf{F}_j\mathbf{F}_i^\top$$

$$= \frac{1}{2}\sum_{i=1}^n \mathbf{F}_i\mathbf{F}_i^\top + \frac{1}{2}\sum_{j=1}^n \mathbf{F}_j\mathbf{F}_j^\top - \sum_{i=1}^n\sum_{j=1}^n \frac{\mathbf{A}_{ij}}{d_i+1}\mathbf{F}_j\mathbf{F}_i^\top$$

$$= \frac{1}{2}\left(\sum_{i=1}^n\sum_{j=1}^n \frac{\mathbf{A}_{ij}\mathbf{F}_i\mathbf{F}_i^\top}{d_i+1} + \sum_{i=1}^n\sum_{j=1}^n \frac{\mathbf{A}_{ij}\mathbf{F}_j\mathbf{F}_j^\top}{d_i+1} - 2\sum_{i=1}^n\sum_{j=1}^n \frac{\mathbf{A}_{ij}}{\sqrt{d_i+1}\sqrt{d_i+1}}\mathbf{F}_j\mathbf{F}_i^\top\right) \text{ undirected graph}$$

$$= \frac{1}{2}\left(\sum_{i=1}^n\sum_{j=1}^n \mathbf{A}_{ij}\left(\frac{\mathbf{F}_i}{\sqrt{d_i+1}} - \frac{\mathbf{F}_j}{\sqrt{d_i+1}}\right)\left(\frac{\mathbf{F}_i^\top}{\sqrt{d_i+1}} - \frac{\mathbf{F}_j^\top}{\sqrt{d_i+1}}\right)\right)$$

$$= \frac{1}{2}\left(\sum_{i=1}^n\sum_{j=1}^n \mathbf{A}_{ij}\left\|\frac{\mathbf{F}_i}{\sqrt{d_i+1}} - \frac{\mathbf{F}_j}{\sqrt{d_i+1}}\right\|_2^2\right) = \sum_{(i,j)\in\mathcal{E}} \mathbf{A}_{ij}\left\|\frac{\mathbf{F}_i}{\sqrt{d_i+1}} - \frac{\mathbf{F}_j}{\sqrt{d_i+1}}\right\|_2^2.$$

# G    DATASETS DETAILS

Cora, Citeseer, and Pubmed are standard citation network benchmark datasets (Sen et al., 2008). Coauthor-CS and Coauthor-Phy are extracted from Microsoft Academic Graph (Shchur et al., 2018). Cornell, Texas, Wisconsin, and Actor are constructed by Pei et al. (2020). ogbn-products is a large-scale product, constructed by Hu et al. (2020).

Table 6: Datasets statistics

| Dataset | # Nodes | # Edges | # Features | # Classes |
|---|---|---|---|---|
| Cora | 2708 | 5429 | 1433 | 7 |
| Citeseer | 3327 | 4732 | 3703 | 6 |
| Pubmed | 19717 | 44338 | 500 | 3 |
| Cornell | 183 | 295 | 1703 | 5 |
| Texas | 183 | 309 | 1703 | 5 |
| Wisconsin | 251 | 499 | 1703 | 5 |
| Actor | 7600 | 33544 | 931 | 5 |
| Coauthor-CS | 18333 | 81894 | 6805 | 15 |
| Coauthor-Phy | 34493 | 247962 | 8415 | 5 |
| ogbn-products | 2449029 | 61859140 | 100 | 42 |

# H    REPRODUCIBILITY

## H.1    IMPLEMENTATION DETAILS

We use Pytorch (Paszke et al., 2019) and PyG (Fey & Lenssen, 2019) to implement NGC and RNGC. The codes of baselines are implemented referring to the implementation of MLP[7][8], GCN[9][10], GAT[11], GLP[12], S$^2$GC[13], and IRLS[14]. All the experiments in this work are conducted on a single NVIDIA Tesla A100 with 80GB memory size. The software that we use for experiments are Python 3.6.8, pytorch 1.9.0, pytorch-scatter 2.0.9, pytorch-sparse 0.6.12, pyg 2.0.3, ogb 1.3.4, numpy 1.19.5, torchvision 0.10.0, and CUDA 11.1.

## H.2    HYPERPARAMETER DETAILS

We provide details about hyparatemeters of NGC and RNGC in Table 7, 8, 9, 10, and 11.

# I    ADDITIONAL EXPERIMENTS

## I.1    ANALYSIS ON ROW NORMALIZATION

In this section, we analyze the influence of row normalization on denoising performance. The noise level $\xi$ controls the magnitude of the Gaussian noise we add to the feature matrix: $\mathbf{X} + \xi\boldsymbol{\eta}$ where $\boldsymbol{\eta}$ is sampled from standard i.i.d., Gaussian distribution. For Cora, Citeseer, and Pubmed, we test $\xi \in \{1, 10, 100\}$. From Table 12, we can observe that the denoising performance of w/ row normalization is better than w/o row normalization. Since row normalization can shrink the value

---

[7]https://github.com/tkipf/pygcn

[8]https://github.com/snap-stanford/ogb/blob/master/examples/nodeproppred/products/mlp.py

[9]https://github.com/tkipf/pygcn

[10]https://github.com/snap-stanford/ogb/blob/master/examples/nodeproppred/products/gnn.py

[11]https://github.com/pyg-team/pytorch_geometric/blob/master/examples/gat.py

[12]https://github.com/liqimai/Efficient-SSL

[13]https://github.com/allenhaozhu/SSGC

[14]https://github.com/FFTYYY/TWIRLS

Table 7: The hyper-parameters for NGC and RNGC on three citation datasets.

| Model | dataset | runs | lr | epochs | wight decay | hidden | dropout | $S$ | $\lambda$ | $\epsilon$ |
|---|---|---|---|---|---|---|---|---|---|---|
| NGC | Cora | 100 | 0.2 | 100 | 1e-5 | 0 | 0 | 16 | 32 | - |
| NGC | Citeseer | 100 | 0.2 | 100 | 1e-5 | 0 | 0 | 16 | 32 | - |
| NGC | Pubmed | 100 | 0.2 | 100 | 1e-5 | 0 | 0 | 16 | 32 | - |
| RNGC | Cora | 100 | 0.2 | 100 | 1e-5 | 0 | 0 | 16 | 32 | 1 |
| RNGC | Citeseer | 100 | 0.2 | 100 | 1e-5 | 0 | 0 | 16 | 32 | 1 |
| RNGC | Pubmed | 100 | 0.2 | 100 | 1e-5 | 0 | 0 | 16 | 32 | 1 |

Table 8: The hyper-parameters for NGC and RNGC on four heterophily graphs.

| Model | dataset | noise level | runs | lr | epochs | wight decay | hidden | dropout | $S$ | $\lambda$ | $\epsilon$ | +MLP |
|---|---|---|---|---|---|---|---|---|---|---|---|---|
| NGC | Cornell | 0.01 | 10 | 0.2 | 200 | 5e-4 | 16 | 0.5 | 16 | 1 | - | y |
| NGC | Cornell | 1 | 10 | 0.2 | 200 | 5e-4 | 16 | 0.5 | 16 | 1024 | - | y |
| NGC | Texas | 0.01 | 10 | 0.2 | 200 | 5e-4 | 16 | 0.5 | 16 | 1 | - | y |
| NGC | Texas | 1 | 10 | 0.2 | 200 | 5e-4 | 16 | 0.5 | 16 | 1024 | - | y |
| NGC | Wisconsin | 0.01 | 10 | 0.2 | 1000 | 5e-4 | 16 | 0.5 | 2 | 1 | - | y |
| NGC | Wisconsin | 1 | 10 | 0.2 | 1000 | 5e-4 | 16 | 0.5 | 2 | 1024 | - | y |
| NGC | Actor | 0.01 | 10 | 0.2 | 1000 | 5e-4 | 16 | 0.5 | 2 | 1 | - | y |
| NGC | Actor | 1 | 10 | 0.2 | 1000 | 5e-4 | 16 | 0.5 | 2 | 1024 | - | y |
| RNGC | Cornell | 0.01 | 10 | 0.2 | 200 | 5e-4 | 16 | 0.5 | 16 | 1 | 1 | y |
| RNGC | Cornell | 1 | 10 | 0.2 | 200 | 5e-4 | 16 | 0.5 | 16 | 1024 | 1 | y |
| RNGC | Texas | 0.01 | 10 | 0.2 | 200 | 5e-4 | 16 | 0.5 | 16 | 1 | 1 | y |
| RNGC | Texas | 1 | 10 | 0.2 | 200 | 5e-4 | 16 | 0.5 | 16 | 1024 | 1 | y |
| RNGC | Wisconsin | 0.01 | 10 | 0.2 | 1000 | 5e-4 | 16 | 0.5 | 2 | 1 | 1e-5 | y |
| RNGC | Wisconsin | 1 | 10 | 0.2 | 1000 | 5e-4 | 16 | 0.5 | 2 | 1024 | 1e-5 | y |
| RNGC | Actor | 0.01 | 10 | 0.2 | 1000 | 5e-4 | 16 | 0.5 | 2 | 1 | 1e-5 | y |
| RNGC | Actor | 1 | 10 | 0.2 | 1000 | 5e-4 | 16 | 0.5 | 2 | 1024 | 1e-5 | y |

of elements in $\boldsymbol{\eta}$, thus reducing the variance $\sigma$. In other words, row normalization make $\left\|\widetilde{\boldsymbol{\mathcal{A}}}_S \boldsymbol{\eta}\right\|_F^2$ converge to zero faster.

## I.2 ANALYSIS ON THE DEPTH OF NGC AND RNGC

In this section, we analyze the influence of the depth of NGC and RNGC model on denoising performance by testing the classification accuracy on semi-supervised node classification tasks. We conduct two sets of experiments: with/without noise in feature matrix. For experiment with feature noise, we simple fix the noise level $\xi = 1$. In each set of experiments, we evaluate the test accuracy with respect to NGC and RNGC model depth, which corresponding to the value of $S$ in $\widetilde{\mathcal{A}}_S$. From Figure 4 and 5, we can observe that the test accuracy barely changes with depth if the model is trained on the clean features on Cora and Pubmed but changes greatly if the model is trained on the clean feature on Citeseer. In this regard, the over-smoothing issue exists in RNGC model on citeseer. However, the denoising performance of shallow RNGC is not good as deeper RNGC models, especially on the large graph like Pubmed. This suggests that we do need to increase the depth of GNN model to include more higher-order neighbors for better denoising performances.

## I.3 MORE DETAILS ABOUT THE STD INFORMATION OF THE RESULTS IN TABLE 4 AND 5

Table 9: The hyper-parameters for NGC and RNGC on two co-author datasets.

| Model | dataset | noise level | runs | lr | epochs | wight decay | hidden | dropout | $S$ | $\lambda$ | $\epsilon$ |
|---|---|---|---|---|---|---|---|---|---|---|---|
| NGC | Coauthor-CS | 0.1 | 10 | 0.2 | 1000 | 1e-7 | 0 | 0 | 16 | 1 | - |
| NGC | Coauthor-CS | 1 | 10 | 0.2 | 1000 | 1e-7 | 0 | 0 | 16 | 128 | - |
| NGC | Coauthor-Phy | 0.1 | 10 | 0.2 | 200 | 5e-4 | 16 | 0.5 | 16 | 1 | - |
| NGC | Coauthor-Phy | 1 | 10 | 0.2 | 200 | 5e-4 | 16 | 0.5 | 16 | 1024 | - |
| RNGC | Coauthor-CS | 0.1 | 10 | 0.2 | 1000 | 1e-7 | 0 | 0 | 16 | 1 | 1 |
| RNGC | Coauthor-CS | 1 | 10 | 0.2 | 1000 | 1e-7 | 0 | 0 | 16 | 128 | 1 |
| RNGC | Coauthor-Phy | 0.1 | 10 | 0.2 | 200 | 5e-4 | 16 | 0.5 | 16 | 1 | 1 |
| RNGC | Coauthor-Phy | 1 | 10 | 0.2 | 200 | 5e-4 | 16 | 0.5 | 16 | 1024 | 1 |

Table 10: The hyper-parameters for NGC and RNGC on ogbn-products dataset.

| Model | noise level | runs | lr | epochs | hidden | dropout | $S$ | $\lambda$ | $\epsilon$ | layers | +MLP |
|---|---|---|---|---|---|---|---|---|---|---|---|
| NGC | 0.1 | 10 | 0.01 | 300 | 256 | 0.5 | 128 | 32 | - | 3 | y |
| NGC | 1 | 10 | 0.01 | 300 | 256 | 0.5 | 128 | 256 | - | 3 | y |
| RNGC | 0.1 | 10 | 0.01 | 300 | 256 | 0.5 | 128 | 32 | 1e-2 | 3 | y |
| RNGC | 1 | 10 | 0.01 | 300 | 256 | 0.5 | 128 | 256 | 1e-2 | 3 | y |

Table 11: The hyper-parameters for NGC and RNGC on three citation datasets of the flipping experiments.

| Model | dataset | flip probability | runs | lr | epochs | wight decay | hidden | dropout | $S$ | $\lambda$ | $\epsilon$ |
|---|---|---|---|---|---|---|---|---|---|---|---|
| NGC | Cora | 0.1 | 100 | 0.2 | 100 | 1e-5 | 0 | 0 | 32 | 64 | - |
| NGC | Cora | 0.2 | 100 | 0.2 | 100 | 1e-5 | 0 | 0 | 16 | 32 | - |
| NGC | Cora | 0.4 | 100 | 0.2 | 100 | 1e-5 | 0 | 0 | 16 | 32 | - |
| NGC | Citeseer | 0.1 | 100 | 0.2 | 100 | 1e-5 | 0 | 0 | 16 | 32 | - |
| NGC | Citeseer | 0.2 | 100 | 0.2 | 100 | 1e-5 | 0 | 0 | 16 | 32 | - |
| NGC | Citeseer | 0.4 | 100 | 0.2 | 100 | 1e-5 | 0 | 0 | 16 | 32 | - |
| NGC | Pubmed | 0.1 | 100 | 0.2 | 100 | 1e-5 | 0 | 0 | 16 | 32 | - |
| NGC | Pubmed | 0.2 | 100 | 0.2 | 100 | 1e-5 | 0 | 0 | 16 | 32 | - |
| NGC | Pubmed | 0.4 | 100 | 0.2 | 100 | 1e-5 | 0 | 0 | 16 | 32 | - |
| RNGC | Cora | 0.1 | 100 | 0.2 | 100 | 1e-5 | 0 | 0 | 32 | 64 | 1e-5 |
| RNGC | Cora | 0.2 | 100 | 0.2 | 100 | 1e-5 | 0 | 0 | 16 | 32 | 1e-5 |
| RNGC | Cora | 0.4 | 100 | 0.2 | 100 | 1e-5 | 0 | 0 | 16 | 32 | 1e-1 |
| RNGC | Citeseer | 0.1 | 100 | 0.2 | 100 | 1e-5 | 0 | 0 | 16 | 32 | 1e-5 |
| RNGC | Citeseer | 0.2 | 100 | 0.2 | 100 | 1e-5 | 0 | 0 | 16 | 32 | 1e-5 |
| RNGC | Citeseer | 0.4 | 100 | 0.2 | 100 | 1e-5 | 0 | 0 | 16 | 32 | 1e-5 |
| RNGC | Pubmed | 0.1 | 100 | 0.2 | 100 | 1e-5 | 0 | 0 | 16 | 32 | 1e-1 |
| RNGC | Pubmed | 0.2 | 100 | 0.2 | 100 | 1e-5 | 0 | 0 | 16 | 32 | 1e-1 |
| RNGC | Pubmed | 0.4 | 100 | 0.2 | 100 | 1e-5 | 0 | 0 | 16 | 32 | 1e-1 |

Table 12: Summary of results of NGC w/o raw normalization on three datasets in terms of classification accuracy (%)

| Noise Level | Cora | | | Citeseer | | | Pubmed | | |
|---|---|---|---|---|---|---|---|---|---|
| | 1 | 10 | 100 | 1 | 10 | 100 | 1 | 10 | 100 |
| w/o RN | 68.3 | 59.7 | 56.1 | 43.5 | 40.4 | 37.6 | 43.1 | 38.8 | 37.4 |
| w RN | 66.1 | 65.5 | 66.2 | 45.3 | 45.1 | 44.8 | 62.3 | 62.7 | 62.1 |

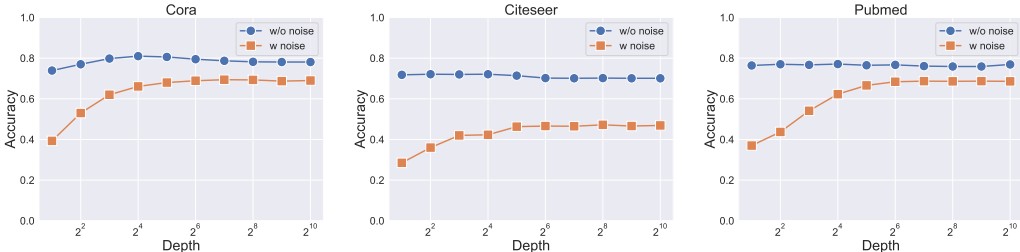

Figure 4: Comparison of classification accuracy v.s. NGC model depth on semi-supervised node classification tasks. The experiments are conducted on clean and noisy features.

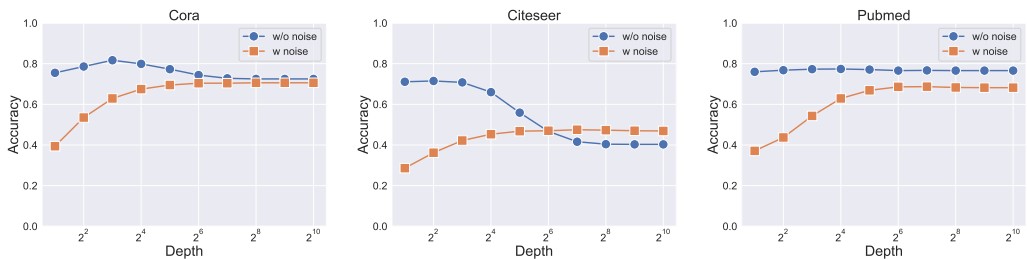

Figure 5: Comparison of classification accuracy v.s. RNGC model depth on semi-supervised node classification tasks. The experiments are conducted on clean and noisy features.

Table 13: Denoising performance over 100 runs against flipping perturbation

| Flipping probability | Cora | | | Citeseer | | | Pubmed | | |
|---|---|---|---|---|---|---|---|---|---|
| | 0.1 | 0.2 | 0.4 | 0.1 | 0.2 | 0.4 | 0.1 | 0.2 | 0.4 |
| MLP | $21.2_{\pm7.3}$ | $21.1_{\pm8.0}$ | $23.3_{\pm8.0}$ | $19.3_{\pm3.3}$ | $18.9_{\pm2.8}$ | $18.9_{\pm2.7}$ | $38.0_{\pm6.3}$ | $39.0_{\pm4.7}$ | $40.6_{\pm2.8}$ |
| GCN | $22.9_{\pm13.6}$ | $19.0_{\pm9.4}$ | $19.0_{\pm9.3}$ | $18.6_{\pm3.4}$ | $18.6_{\pm3.1}$ | $18.5_{\pm3.2}$ | $37.8_{\pm6.6}$ | $38.1_{\pm7.1}$ | $37.6_{\pm8.0}$ |
| GAT | $70.1_{\pm1.5}$ | $65.6_{\pm1.5}$ | $60.0_{\pm3.2}$ | $45.3_{\pm2.9}$ | $39.3_{\pm3.4}$ | $26.0_{\pm5.1}$ | $43.3_{\pm2.7}$ | $49.5_{\pm3.2}$ | $60.0_{\pm4.1}$ |
| GLP | $32.3_{\pm0.8}$ | $30.8_{\pm4.0}$ | $29.0_{\pm6.4}$ | $19.7_{\pm2.7}$ | $18.9_{\pm2.4}$ | $18.8_{\pm2.3}$ | $42.1_{\pm2.0}$ | $41.5_{\pm1.8}$ | $40.7_{\pm0.1}$ |
| $S^2$GC | $75.0_{\pm1.6}$ | $71.5_{\pm2.0}$ | $63.8_{\pm4.4}$ | $49.9_{\pm3.9}$ | $46.4_{\pm3.2}$ | $43.4_{\pm2.9}$ | $50.4_{\pm2.2}$ | $60.2_{\pm1.9}$ | $69.3_{\pm1.6}$ |
| IRLS | $66.4_{\pm2.0}$ | $61.0_{\pm1.9}$ | $54.7_{\pm2.5}$ | $50.3_{\pm2.7}$ | $45.9_{\pm2.1}$ | $43.8_{\pm1.7}$ | $51.4_{\pm4.1}$ | $60.0_{\pm3.8}$ | $69.0_{\pm2.4}$ |
| NGC | $\mathbf{77.5}_{\pm\mathbf{1.3}}$ | $\mathbf{75.3}_{\pm\mathbf{1.3}}$ | $\mathbf{65.7}_{\pm\mathbf{5.6}}$ | $\mathbf{54.9}_{\pm\mathbf{2.6}}$ | $\mathbf{51.9}_{\pm\mathbf{2.7}}$ | $\mathbf{48.5}_{\pm\mathbf{2.4}}$ | $\mathbf{53.0}_{\pm\mathbf{2.1}}$ | $\mathbf{62.3}_{\pm\mathbf{1.6}}$ | $\mathbf{70.4}_{\pm\mathbf{1.1}}$ |
| RNGC | $\mathbf{77.6}_{\pm\mathbf{1.2}}$ | $\mathbf{75.2}_{\pm\mathbf{1.4}}$ | $\mathbf{72.8}_{\pm\mathbf{1.4}}$ | $\mathbf{55.0}_{\pm\mathbf{3.0}}$ | $\mathbf{51.8}_{\pm\mathbf{2.4}}$ | $\mathbf{48.7}_{\pm\mathbf{2.4}}$ | $\mathbf{54.3}_{\pm\mathbf{2.0}}$ | $\mathbf{63.9}_{\pm\mathbf{1.6}}$ | $\mathbf{71.6}_{\pm\mathbf{1.1}}$ |

Table 14: Defense performance over 100 runs against structure attack

| Model | Cora | Citeseer | Pubmed |
|---|---|---|---|
| GCN | $47.53_{\pm1.96}$ | $56.94_{\pm2.09}$ | $75.50_{\pm0.17}$ |
| GAT | $54.78_{\pm0.74}$ | $61.85_{\pm1.12}$ | $65.41_{\pm0.77}$ |
| RobustGCN | $50.51_{\pm0.78}$ | $55.35_{\pm0.66}$ | $67.95_{\pm0.15}$ |
| GCN-Jaccard | $60.82_{\pm1.08}$ | $59.89_{\pm1.47}$ | $83.66_{\pm0.06}$ |
| GCN-SVD | $52.06_{\pm1.19}$ | $57.18_{\pm1.87}$ | $82.72_{\pm0.18}$ |
| $S^2$GC | $51.60_{\pm0.05}$ | $54.11_{\pm0.09}$ | $64.04_{\pm0.03}$ |
| RNGC | $63.16_{\pm1.71}$ | $65.64_{\pm1.76}$ | $84.04_{\pm0.25}$ |
| ProGNN | $\mathbf{69.72}_{\pm\mathbf{1.69}}$ | $\mathbf{68.95}_{\pm\mathbf{2.78}}$ | $\mathbf{86.76}_{\pm\mathbf{0.19}}$ |

