# OpenReview forum: "How Powerful is Implicit Denoising in Graph Neural Networks"
_ICLR.cc/2023/Conference — Submitted to ICLR 2023_

### Official Review · Reviewer_2Src · 2022-10-24

**Confidence:** 5
**Correctness:** 4
**Technical Novelty And Significance:** 4
**Empirical Novelty And Significance:** 4
**Recommendation:** 6

**Clarity, Quality, Novelty And Reproducibility:**

The quality, clarity, and originality of this work are generally above the average bar of ICLR publication.

**Strength And Weaknesses:**

Strengths:
+ The readability of this work is good. It is easy to follow since the motivation and presentation are quite clear;
+ The theoretical formulation and analysis are generally solid;
+ The proposed adversarial graph signal denoising problem is interesting and may potentially attract more researchers who are expecting to develop advanced GNNs via solving the GSD problem.

Weaknesses:
+ The relationship between NGC and the existing multi-scale GNNs is not clear. Some remarks on clarifying this issue are helpful;
+ Based on the experimental results shown in Section 5, the advantages of RNGC over NGC are not significant. More discussion on possible reasons are beneficial for improving the clarity of this work;
+ The footnote of Remark 1 mentions that you adopt another form of normalized adjacency matrix in the experiments, instead of the exact form presented in Remark 1. The influence is not studied/clarified with any empirical results;
+ Although the authors present Figure 1 as a case study on the influence of graph structure on implicit denoising, more details (to be added in the Appendices) are needed to show how the $\tau$ values are calculated. Also, the records for Graph 1~4 are not very convincing to conclude the insight (5), that is, "In terms of graph size, the graph with a larger size has a better denoising effect." Please clarify this issue;
+ To develop RNGC, the authors have used a little trick that "....approximate Eq. (16) by replacing the F with X in the
inverse matrix on the right-hand side...." The rationale should be discussed. Also, how is the convergence property of RNGC?


**Summary Of The Paper:**

This paper studies in-depth the implicit denoising effect in graph neural networks, which remains an open issue from the theoretical perspective. Rigorous theoretical analysis is provided to uncover the underlying philosophy of GNNs for graph signal denoising. The theoretical results and discussion suggest that the implicit denoising effect is related to the graph structure, size, and the GNN model's architecture. A robust GNN model is developed by solving a novel adversarial graph signal denoising probelm. Extensive experimental studies are presented to verify the obtained theoretical results and the effectiveness of the proposed robust GNN model.

**Summary Of The Review:**

Overall, I vote for a weak accepting. I like the idea of studying the implicit denoising effect via defining and analyzing the high-order graph connectivity factor. My major concern is about the weaknesses raised above. Hopefully, the authors can address my concern in the rebuttal period.

---

> ### Author Response · Authors · 2022-11-14
> **Response to Reviewer 2Src**
>
> **Q1:** The relationship between NGC and the existing multi-scale GNNs is not clear.
> **A1:** We are sorry for the confusion. Multi-scale GCN combines information from neighbors with various distances. Specifically, the N-GCN extends the feature aggregation module in GCN by concatenating neighboring feature information using $\hat{A}^k$ at each layer, while for NGC, we incorporate such multi-scale information by summing up $\left(\frac{\lambda}{\lambda+1} \widetilde{\mathcal{A}}\right)^s$ as can be seen from Eq. (4). We have revised the paper to discuss the difference.
>
> **Q2:** The advantages of RNGC over NGC are not significant.
> **A2:** Please note that RNGC is just an extension of our NGC framework which also enjoys decent denoising performances while our main contribution still lies in the theoretical analysis of the denoising problem. Indeed the performance gain of RNGC compared with NGC is not that significant, yet in general, RNGC still slightly outperforms NGC, which suggests that adversarial training style augmentation could also be helpful in denoising problem.
>
> **Q3:** The influence from another normalized adjacency matrix is not studied.
> **A3:** Thank you for your suggestion. We further conduct experiments for NGC under the other normalized adjacency matrix $\widetilde{\mathbf{\mathcal{A}}}=\widetilde{\mathbf{D}}^{-1}\widetilde{\mathbf{A}}$. The experimental results are as follows. Note that the NGC (S) is for $\widetilde{\mathcal{A}}=\widetilde{\mathbf{D}}^{-\frac{1}{2}} \widetilde{\mathbf{A}} \widetilde{\mathbf{D}}^{-\frac{1}{2}}$, and NGC (R) is for $\widetilde{\mathbf{\mathcal{A}}}=\widetilde{\mathbf{D}}^{-1}\widetilde{\mathbf{A}}$.
>
> |  |  |  | Cora |  |  |
> | :---: | :---: | :---: | :---: | :---: | :---: |
> | Noise Level | 0.1 | 0.2 | 0.3 | 0.4 | 0.5 |
> | NGC (S) | 76.37+-3.23 | 71.74+-3.39 | 68.56+-4.17 | 67.03+-4.25 | 66.53+-5.14 |
> | NGC (R) | 76.34+-2.90 | 72.13+-3.03 | 68.90+-3.97 | 67.74+-4.50 | 67.28+-4.74 |
>
> **Q4:** How the $\tau$ values are calculated?
> **A4:** We are sorry for this. We calculate $\tau$ by definition, i.e., using Eq. (5).
>
> **Q5:** Clarify "the graph with a larger size has a better denoising effect".
> **A5:** Our Theorem 1 suggests that to obtain the best denoising performance, we should have smaller $\tau$ and larger $n$. $\tau$ is largely determined by the graph structure/connectivity (see Eq. (5)) while $n$ is the number of nodes on a graph (graph size). If two graphs have similar graph structures (i.e., same $\tau$) but with a larger $n$, we would have a better denoising effect according to our Theorem 1.
>
> **Q6:** The rationale to use the trick to obtain RNGC/How is the convergence property of RNGC?
> **A6:** The approximation is needed since it is difficult to get the close-form of $\mathbf{F}$ by solving Eq. (16). Note that in Eq. (14) we also require $\mathbf{F}$ to be close to $\mathbf{X}$ do to the first term. Therefore, we can approximate Eq. (16) by replacing the $\mathbf{F}$ with $\mathbf{X}$ in the inverse matrix on the right hand side. Due to this approximation, we are not able to exactly analyze the convergence property of RNGC from the theoretical perspective, yet empirically, we observe very similar convergence curves of RNGC similar to NGC.
>
> [1] N-GCN: Multi-scale Graph Convolution for Semi-supervised Node Classification.

---

> ### Author Response · Authors · 2022-12-09
> **We hope the reviewer find our response helpful**
>
> We appreciate the reviewer for providing many insightful feedbacks, which greatly help us improve the clarity and scope of our paper. We tried to provide a thorough response and additional results. We sincerely hope the reviewer finds our response useful and updates the scores if your concerns have been resolved. We are also open to further discussion if there are further questions.

---

### Official Review · Reviewer_Jway · 2022-10-24

**Confidence:** 3
**Correctness:** 2
**Technical Novelty And Significance:** 2
**Empirical Novelty And Significance:** 1
**Recommendation:** 3

**Clarity, Quality, Novelty And Reproducibility:**


Clarity:
- paper is not well written. More importantly, some sentences are incomplete:
> The relationship between GSD and GCN can be briefly illustrated as follows (Ma
et al., 2021).
- some important quantities are not defined: for instance, In Eq (1) and (2), lambda is not defined. I could try to guess it's like a Lagrange multiplier, but it's not obvious.  It is set to 64 in Fig. 1
- In Eq. (31), d does not appear.
- some parts are especially unclear, like the top of page 6.
- remark 3 refers to sec 3 (twice), which it is in.. ?

Quality:
- discussion of denoising vs oversmoothing compromise is missing, making the point of the whole paper unclear.
- many quantities are not defined where they should in equations.
- many arguments are not convincing (see other parts of my review above)
- empirical results are not convincing, albeit rather well reported (error bars are missing in table 5).

Novelty:
- as admitted, NGC is not new. RNGC is, but does not significantly outperform.

Reproducibility:
I did not find the link to the code.



**Strength And Weaknesses:**

Strengths:
- The key concepts and intuitions are explained and after a bit of work, are inferred from the paper.
- Figure 1 clearly illustrates the notion of High-order Graph Connectivity Factor, albeit only on very simple graphs.

Weaknesses:
- the main (empirical) claim, that RNGC outperforms others (NGC, mainly) is not backed by evidence: in fig 1 and table 1,2,3,4, they perform as well, or with very similar accuracy, within the error bar of each other.

    In Table 5, NGC's results are not reported (Also Table 4 does not report error bars).

    (it is not highlighted, but is mentioned that "Compared with NGC, the additional computation cost is O(|E|)." (understand: linear in the number of edges). This can be large compared to the number of nodes, in industrial graphs. If it's just a +O(E), it's ok. If it's multiplicative, it is not negligible at all.)

- The theoretical analysis suffers two main problems:
    * it does not convincingly apply to other GNNs than NGC ("Besides, if we remove the non-linear functions in GCN (Kipf & Welling, 2017), it also can be covered by our model.") This is not convincing. Further more, the 4 assumptions (esp. 3 and 4) of Lemma 1 are not convincingly applicable to most GNNs/datasets (at least it's not obvious). This hinders the generality of the theorem 1, which is the theoretical contribution of the work.
    * it does not predict any kind of non-trivial, precise or insightful denoising effects. Authors claim themselves :
      > "We prove that with the increase in graph size and graph connectivity factor, the stochasticity tends to diminish, which is called the “denoising effect” in our work." -> This result is very intuitive and pretty trivial, it seems.
     > "Besides, GNN architectures also affect the convergence rate. Deeper GNNs can have a faster convergence rate." -> this also seems very trivial to me.

     Other than this, the effect of graph structure is hidden in the "High-order Graph Connectivity Factor" \tau, which is not very transparent, but could be argued is an important contribution (it's hard to say, it's not advertized this way)
- The authors fail to discuss the compromise between Denoising and Over-smoothing, although at first sight this seems like a very obvious problem to me ! Then, somehow NGC performs well on heterophilic graphs.. but why, I do not understand. Maybe precisely because NGC does not smooth directly, as a GNN would do ?
- Actually, oversmoothing appears in 4 and 5, hidden in the Appendix: on the original tasks (without added noise), the performance decreases when adding too many layers.


**Summary Of The Paper:**

The paper aims to analyze the Denoising effect provided by GNNs (on corrupted graph data, typically with noisy labels).

It recalls the Neumann Graph Convolution (NGC) perspective and how it connects with Graph Signal Denoising (GSD).

It defines the High-order Graph Connectivity Factor, \tau, which is a measure of the average connectivity of the graph, and of its structure.

It proposes a theoretical estimation of the denoising effect of a GNN, in the NGC framework, in the form of theorem 1: denoising is stronger for larger graphs, and linearly depend on \tau, the above-defined High-order Graph Connectivity Factor.

It proposes an adversarial setup, the adversarial graph signal denoising problem, and its solution, in the form of an algorithm, Robust NGC.

The RNGC algorithm does not significantly outperform the pre-existing NGC approach: average performance is almost always exactly equal, and is always perfectly in the same range of performance (error bars overlapping with average value).


**Summary Of The Review:**

Overall, this paper does not produce what it claims to, both empirically and theoretically.

In addition, the very point of the paper (denoising) is not clearly made (I would expect more denoising also means more oversmoothing, i.e. not always better performance).

---

> ### Author Response · Authors · 2022-11-14
> **Response to Reviewer Jway - Part 1**
>
> Before answering your questions, we want to make a clarification.
> *Reviewer Jway: "The paper aims to analyze the Denoising effect provided by GNNs (on corrupted graph data, typically with noisy labels)."*
> We do not focus on the analysis of denoising effect on corrupted graph data with noisy labels. Instead, we analyze how GNNs eliminate the noise in the feature matrix, which is completely from noisy label settings.
>
> **Q1:** The concern about our claim "RNGC outperforms NGC"
> **A1:** Please note that RNGC is just an extension of our NGC framework which also enjoys decent denoising performances while our main contribution still lies in the theoretical analysis of the denoising problem. Indeed the performance gain of RNGC compared with NGC is not that significant, yet in general, RNGC still slightly outperforms NGC, which suggests that adversarial training style augmentation could also be helpful in denoising problem.
>
> **Q2:** In Table 5, NGC's results are not reported (Also Table 4 does not report error bars).
> **A2:** Due to the limit of the space, we didn't report std and we have reported these in the Appendix in the revised version. Besides, we have emphasized RNGC is related to the graph structure perturbations. So we conduct the experiments of RNGC against graph structure meta-attack. Yet NGC has nothing to do with graph structure perturbation and thus we didn't compare it there.
>
> **Q3:** The computational complexity of RNGC compared with NGC.
> **A3:** We think there is a misunderstanding of the computational complexity of the inner product of feature vectors. Please refer to the details in "The computational complexity of evaluating Eq. (9) is then $\mathcal{O}(|\mathcal{E}| C H F)$, i.e. linear in the number of graph edges." [1]. And we compute the inner product of feature vectors, so the additional computational complexity is $\mathcal{O}(|\mathcal{E}| D)$, where $D$ is the dimension of the feature vector. In other words, the additional computational complexity is the same as with the one-time aggregation of GNNs.
>
> **Q4:** The generality of the our theoretical analysis.
> **A4:** First of all, it would be hard to find a single theorem that could cover all types of GNNs. For theoretical analysis, it is common to target a simple representative type (i.e., graph spectral convolution in our case) for the ease of theoretical analysis. Specifically, our framework directly reflects the role of feature aggregation, a key component for a broad range of GNNs, while simplifying the later nonlinear activation part. Therefore, although our framework targets a simplified version of GCN, it still reflects the key mechanism of GCN and explains why it also enjoys the denoising effect. Thus we believe our theoretical analysis are general enough to provide meaningful insights to the GNN denoising effects.
>
> **Q5:** The 4 assumptions (esp. 3 and 4) of Lemma 1 are not convincingly applicable to most GNNs/datasets.
> **A5:** First of all, Lemma 1 only relies on Assumptions 1 and 2. Assumptions 3 and 4 are also not strong assumptions by any sense. Specifically, Assumption 3 only states that the Frobenius norm of network weights is bounded by constant. This can be easily satisfied if the trained GNN model weights are not outrageously large for all dimensions. Assumption 4 is about the loss function smoothness, which is a common assumption in gradient based convergence analysis. Essentially if the gradient norm is bounded during the training procedure, this assumption is satisfied. While empirically we usually observe that the gradient norm is decreasing along the training trajectory.
>
> **Q6:** The concern that our theorem does not predict any kind of non-trivial, precise or insightful denoising effects.
> **A6:** We are sorry for the confusion. From our Theorem 1, we have shown that the denoising effect is influenced by graph size $n$ and the high-order graph connectivity factor $\tau$, which reflects the graph structure and GNN architecture. Therefore, a larger graph size (large $n$) comes with better denoising performances. Considering the graph connectivity factor $\tau$, we can observe that deeper GNNs (larger $S$) will lead to smaller $\tau$ and thus better denoising performances. This is also verified by the empirical results. Moreover, as demonstrated in the Case Study on page 5, different graph connectivity will also affect the value of $\tau$ (more connected graphs leads to smaller $\tau$ and thus better denoising performances). We have updated the paper to better demonstrate this.

---

> > ### Author Response · Authors · 2022-11-14
> > **Response to Reviewer Jway - Part 2**
> >
> > **Q7:** The effect of graph structure is hidden in the "High-order Graph Connectivity Factor" $\tau$.
> > **A7:** We are sorry for the confusion but we are not trying to hide anything here. The relation of the High-order Graph Connectivity Factor and graph structure is actually easy to understand: more connected graphs lead to smaller $\tau$. This can be easily seen from Eq. (7), the row sum of $[\widetilde{\mathcal{A}}_S]_i^2$ is upper and lower bounded. Considering the extreme case that the graph is fully-connected, this row sum will reach the lower bound, which means $\tau = 1$ by definition of $\tau$. On the other hand, if the graph is completely disconnected, the row sum reaches the upper bound and thus $\tau = n$. Therefore, $\tau$ directly reflects the (high-order) connectivity of the graph.
> >
> > **Q8:** The relationship between over-smoothing and denoising/over-smoothing appears in Figure 4 and 5.
> > **A8:** Thanks for the great question. First, we want to emphasize that our denoising analysis is mainly from an optimization perspective, i.e., with feature noise, will the GNN training procedure still able to converge to a good solution? While over-smoothing is in general referred as a generalization problem, i.e., with a large number of layers and sufficient feature smoothing, the GNN training loss can still be minimized but the test performance is not ideal due to over-smoothed feature representation on test nodes. From this perspective, it is easy to understand why denoising and over-smoothing could co-exist: we only address the optimization problem caused by the noises but not the generalization issue.
> >
> > **Q9:** The concern that why NGC performs well on heterophilic graphs.
> > **A9:** On heterophilic graphs, MLPs are shown to achieve better performances than more GNNs such as GCN since feature aggregation may give inaccurate information. For our NGC method, note that we keep the zero-order term (no aggregation like MLP) with the largest weight and thus preserve the original feature information while also considering high-order terms for better denoising performances. Thus in the context of the denoising problem, the design of NGC actually take the advantage of both zeroth-order and high-order information to achieve better denoising performances.
> >
> > **Q10:** Minor issue: 1) some sentences are incomplete. 2) About the lambda in Eq. (1). 3) The concern that In Eq. (31), $d$ does not appear. 4) The concern that some parts are especially unclear, like the top of page 6. 5)  Minor issue: remark 3 refers to sec 3 (twice), which it is in.. ?
> > **A10:** 1) We thank the reviewer for pointing out this and have removed this sentence. 2) The $\lambda$ controls the $\ell_2$-based graph smoothing as [5] illustrates. 3) We think there is a misunderstanding of our proof. We extend the $f\left(\mathbf{W}_f\right)$ in Eq. (34). So $d$ does not appear in Eq. (31). 4) Please demonstrate your concern more clearly so that we can address your concern. 5) We are sorry for this, and have corrected it in our revised version based on your suggestion.
> >
> > [1] Semi-Supervised Classification with Graph Convolutional Networks.
> >
> > [2] Simple and Deep Graph Convolutional Networks.
> >
> > [3] Training graph neural networks with 1000 layers.
> >
> > [4] Simple Spectral Graph Convolution.
> >
> > [5] Elastic Graph Neural Networks.

---

> ### Comment · Reviewer_Jway · 2022-11-17
> **updated review**
>
>
> Thank you for your detailed answers.
>
> Especially, thank you for table 13, which indeed shows that RNGC does not significantly differ from NGC in terms of performance.
> This implies that the main contribution of the work is the theoretical one, in the view of inspiring GNN architeecture design, for denoising wihtout over-smootthing.
>
>
>
> However, this theoretical/intuition-oriented part of the work is still not fully convincing to me.
> I reckon I am clearnly not familiar with the GSD problem, however I am now quite familiar with GNNs, and I have to say that the supposedly new intuitions provided by the work in this paper are not obvious to me.
> I think the paper would reach its goal if it were truly accessible to the newcomer, provoding simple and non-trivial intuitions about graph denoising.
>
> And ultimately the ideal work would provide tips/guidelines about how to improve GNNs for heterophilic graphs prediction. Given that graph structure is fixed, in a given task, this would mean, providing an interplay between the theory of denoising and GNN design (not graph-data structure, which is fixed).
>
> I change my rating from
> > 1: strong reject
>
> to
>
> > 3: reject, not good enough
>
> and I change from confidence 4 to 3.
>
>
>
> Apart from this:
>
> I agree with the points raised by reviewer ZXuu.
>
> I did not really understand A8:
> > A8: Thanks for the great question. First, we want to emphasize that our denoising analysis is mainly from an optimization perspective, i.e., with feature noise, will the GNN training procedure still able to converge to a good solution? While over-smoothing is in general referred as a generalization problem, i.e., with a large number of layers and sufficient feature smoothing, the GNN training loss can still be minimized but the test performance is not ideal due to over-smoothed feature representation on test nodes. From this perspective, it is easy to understand why denoising and over-smoothing could co-exist: we only address the optimization problem caused by the noises but not the generalization issue.
>
> I think the argument developped in A9, should be included in the paper. currently one has no intuition over this fact.

---

> > ### Author Response · Authors · 2022-11-19
> > **updated response**
> >
> > Thank you for increasing the score. Before answering your questions, we want to make a clarification about the main contribution of our paper.
> >
> > *Reviewer Jway: "This implies that the main contribution of the work is the theoretical one, in the view of inspiring GNN architecture design, for denoising without over-smoothing."*
> >
> > We think there is still a misunderstanding here. The main contribution of our paper is to explain why and when GNN can have such a denoising effect, but not about GNN architecture design. Moreover, please note that we didn't specifically aim at combating "over-smoothing" as we are mainly talking about denoising alone. See more details in the following.
> >
> > **Q11:** Given that graph structure is fixed, in a given task, providing an interplay between the theory of denoising and GNN design.
> > **A11:** Please note that this is not our main focus and contribution as we mainly study why and when denoising happens through the lens of NGC. In fact, it is impossible for us to give your desired "interplay" as our analysis is doing exactly the opposite thing: we are fixing the GNN design to be NGC and study what are factors that may affect its denoising performances, not the other way around. We agree that designing new GNN architectures for denoising sounds interesting and may be our next step, understanding how denoising happens in the current GNN architectures is equally important and is actually the foundation of your desired "interplay". Therefore, we believe our contribution is significant.
> >
> > **Q12:** I agree with the points raised by reviewer ZXuu.
> > **A12:** We have answered the questions raised by reviewer ZXuu. If you have any other questions, please let us know so that we can address your concern.
> >
> > **Q13:** Didn't understand A8.
> > **A13:** As the reviewer didn't specify which part causes the confusion. We can only try to explain the whole thing with more detail.
> > As we all know, the total error caused by the machine learning model can be separated into two parts: optimization error (error caused by not fully optimizing the training loss) and generation error (error caused by the difference between train/test data). Please note our denoising analysis mainly focuses on the optimization part. Essentially, Theorem 1 suggests that with feature noise, the GNN training procedure is still able to converge to a good solution (close to the actual minimizer of training loss). While over-smoothing is in general referring to the large generalization error, i.e., the GNN training loss has been minimized (low training loss) but the test performance is not ideal due to over-smoothed feature representation on test data.  Therefore, while we can make sure that the random noise does not cause troubles for achieving low optimization error, the overs-moothing effect (not ideal generalization error) could still remain.
> >
> > **Q14:** The argument developed in A9 should be included in the paper.
> > **A14:** We thank the reviewer for this suggestion. We have included this argument in the experiment section in the revised version.
> >
> >
> > Furthermore, if you have any question about our paper, please let us know so that we can address your concern in the rebuttal stage.

---

> ### Author Response · Authors · 2022-12-12
> **We hope the reviewer find our response helpful**
>
> We appreciate the reviewer for providing many insightful feedbacks, which greatly help us improve the clarity and scope of our paper. We tried to provide a thorough response and additional results. We sincerely hope the reviewer finds our response useful and updates the scores if your concerns have been resolved. We are also open to further discussion if there are further questions.

---

### Official Review · Reviewer_ZXuu · 2022-10-25

**Confidence:** 4
**Correctness:** 3
**Technical Novelty And Significance:** 2
**Empirical Novelty And Significance:** 3
**Recommendation:** 5

**Clarity, Quality, Novelty And Reproducibility:**

The paper is clearly written. The novelty is good but lacks the right intuition and convincing experiments.

**Strength And Weaknesses:**

# Strength:

1. The paper is clearly written and easy to follow.

2. The paper provides a theoretical analysis of the denoising effect of graph convolution. The impact of graph size, connectivity, expansion order are discussed. This further strengthens previous unified understanding of designing GNNs from the graph signal denoising perspective.

3. The paper proposes a robust Neumann graph convolution model inspired by adversarial training. This perspective is novel and interesting (but with many concerns as discussed below).



# Weakness：

1. The theoretical analysis is based on a specific and simplified GNN architecture (a linear layer followed by a diffusion approximated by a Naumann series). This does not align with classic and popular GNN architectures and therefore might not correctly reveal the denoising effects of GNNs. Moreover, the problem being analysed in Section 3 (Eq. (8)) is a fully-supervised problem with all nodes being labeled data. It is unclear how it aligns with the common semi-supervised setting in GNNs.

2. I believe there is much literature about graph signal processing with statistical analysis on noise, but the paper does not cite any of them. It will be better to discuss how the theoretical results improve what has been done in the literature if there are any.

3. The is a lack of right intuition in the proposed robust Neumann graph convolution model (RNGC). In the definition of the adversarial graph signal denoising problem, the adversary tries to perturb the graph which enlarges the distance between connected neighbors. In other words, in the forward computation of RNGC, the graph is perturbed such that dissimilar nodes are connected as shown in Eq. (17). Intuitively, this will have a negative impact on the final performance even in the clean feature setting. Furthermore, the noise in the feature might make similar nodes (measured by original clean features) dissimilar (measured by noise features), which makes it even worse. It is unclear how the proposed formulation can help mitigate the impact of noise features (or noise graphs).

4. The theoretical analysis does not provide new insight into how to design better GNNs. In fact, the analysis is irrelevant to the proposed idea in Section 4.

5. The experimental results are not convincing.

(1) First of all, the comparison between the proposed model and the baselines is unfair. For instance, it is mentioned in Section 5 that GCN and GAT have 2 layers, IRLS has 8 layers (or more but unclear), while NGC and RNGC have 16 layers. Since more layers might achieve better denoising effect, such a comparison can not justify the advantages of the proposed models.

(2) Moreover, the baselines are quite weak. I wonder how APPNP [1] and AirGNN [2] perform in the considered settings with a fair number of propagation layers. APPNP is a GNN that exactly follows the graph denoising problem in Eq. (1). AirGNN is a GNN specifically designed for handling noise features with a new graph denoising problem. In terms of graph structure attacks, ProGNN [3] and many others usually provide much stronger baselines than those being compared in the paper. It will be more convincing to include this baseline as well.


[1] Predict then Propagate: Graph Neural Networks meet Personalized PageRank, ICLR 2019
[2] Graph Neural Networks with Adaptive Residual, NeurIPS 2021

**Summary Of The Paper:**

The paper provides an analysis of the denoising effect of GNNs and reveals the impact of graph size, connectivity, and GNN architectures. It also proposes a robust Neumann graph convolution (RNGC) model based on the defined adversarial graph signal denoising problem.

**Summary Of The Review:**

The paper analyzes the denoising effect of graph convolutions and proposes a robust Neumann graph convolution inspired by adversarial training. The proposed idea lacks the right intuition and more convincing experiments are needed.

-------
After rebuttal, I increase my score to 5.

---

> ### Author Response · Authors · 2022-11-14
> **Response to Reviewer ZXuu-Part 1**
>
> **Q1:** The theoretical analysis is based on a specific and simplified GNN architecture and does not align with classic and popular GNN architectures.
> **A1:** First of all, it would be hard to find a single theorem that could cover all types of GNNs. For theoretical analysis, it is common to target a simple representative type (i.e., graph spectral convolution in our case) for the ease of theoretical analysis. Specifically, our framework directly reflects the role of feature aggregation, a key component for a broad range of GNNs, while simplifying the later nonlinear activation part. Therefore, although our framework targets a simplified version of GCN, it still reflects the key mechanism of GCN and explains why it also enjoys the denoising effect.
>
> **Q2:** The problem which is analyzed is a fully-supervised problem. It is unclear how it aligns with the common semi-supervised setting.
> **A2:** We are sorry for the confusion. In fact, our framework is general and can be easily extended to semi-supervised settings. Specifically, we can rewrite Eq. (12) as $g(\mathbf{W})=||(\tilde{\mathcal{A}}_S\mathbf{X}^*\mathbf{W})_T-\mathbf{Y}_T||_F^2$, where the $T$ denotes the training samples under the full-supervised setting. Such modifications would lead to the same conclusion in the semi-supervised setting.
>
> **Q3:** The concern about the literature.
> **A3:** Thanks a lot for pointing out this. We have cited and discussed the literature [1, 2] about graph signal processing in the revision (Section 6). We also further discussed more works conducting statistical analysis on the graph noise from the empirical perspective [3-8] in the revised version.
>
> **Q4:** The intuition in the proposed robust Neumann graph convolution model.
> **A4:** The proposed adversarial graph signal denoising problem (AGSD) Eq. (15) follows a min-max optimization formulation. The intuition behind Eq. (15) is similar to adversarial training in traditional robustness studies: we consider the worst-case perturbations on the graph and we hope that if our method could resist worst-case perturbations on the graph, it would be more robust against random noisy inputs. Indeed such a design would cause a certain level of accuracy loss, yet we want to argue that in terms of denoising performances, it achieve better result, as can be shown in Figure 2.
>
> **Q5:** New insight on the design of GNNs.
> **A5:** First we want to emphasize that our main contribution is the theoretical analysis, not designing new architectures. Yet we believe there are new insights drawn from our analysis. From our Theorem 1, we have shown that the denoising effect is influenced by graph size $n$ and the high-order graph connectivity factor $\tau$, which reflects the graph structure and GNN architecture. Considering how the GNN architecture will affect the high-order graph connectivity factor $\tau$, we can conclude that deeper GNNs (larger $S$) will lead to better denoising performances. This is also verified by the empirical results. Note that it is well-known that deeper GNNs would suffer from over-smoothing problems in generalization performances. So how to design a deep GNN model which balances between over-smoothing and denoising is the key for empirical success.
>
> **Q6:** Comparison of baselines with different number of layers.
> **A6:** We understand the reviewer's concern. Yet please note that our main contribution of this paper lies in the theoretical analysis of the graph denoising problem (not building a better GNN architecture) and thus our experimental focus is on verifying whether those factors indeed influence model denoising. As shown in the experimental results, in general, deeper GNNs have better denoising performance compared with shallow GNNs. Also, the denoising performance of GNNs on Pubmed is better than Citeseer, suggesting the power of different graph connectivities. If the reviewer wants to see how the number of layers affects denoising, please see our ablation study in Figures 4 and 5. We didn't put the comparison of different models with the same number of layers since it simply shifts the focus to the superiority of different architectures (as different architectures lead to different representation power and thus different accuracies), which is not our main focus.
>
> **Q7:** Comparison with new baselines.
> **A7:** We provide the experimental results of APPNP and AirGNN as follows:
> |  |  |  | Cora |  |  |
> | :---: | :---: | :---: | :---: | :---: | :---: |
> | Noise Level | 0.1 | 0.2 | 0.3 | 0.4 | 0.5 |
> | APPNP | 78.14+-8.74 | 72.19+-10.25 | 66.34+-12.38 | 62.33+-12.88 | 62.98+-11.79 |
> | AirGNN | 77.72+-11.73 | 72.78+-9.64 | 65.64+-13.02 | 62.33+-18.26 | 62.98+-17.66 |
> | NGC | 76.37+-3.23 | 71.74+-3.39 | 68.56+-4.17 | 67.03+-4.25 | 66.53+-5.14 |
> | RNGC | 77.40+-2.55 | 72.65+-2.76 | 69.11+-3.22 | 67.96+-3.59 | 67.62+-3.33 |

---

> > ### Author Response · Authors · 2022-11-14
> > **Response to Reviewer ZXuu-Part 2**
> >
> > **A7:**  As the review suggests, we have added the results of ProGNN [8] in the revised version. The performance is higher than ours. Yet please note that our paper only focused on removing random noise in features, not against adversarial attacks. The comparison with those defense baselines is only to show the unexpected performance on defending attacks.
> >
> > [1] Graph denoising with framelet regularizer.
> >
> > [2] Signal denoising on graphs via graph filtering.
> >
> > [3] Graph Neural Networks with Adaptive Residual.
> >
> > [4] Graph Trend Filtering Networks for Recommendation.
> >
> > [5] Is Homophily a Necessity for Graph Neural Networks?
> >
> > [6] Graph Feature Gating Networks.
> >
> > [7] Elastic Graph Neural Networks.
> >
> > [8] Graph structure learning for robust graph neural networks.

---

> > ### Comment · Reviewer_ZXuu · 2022-11-22
> > **Further comments**
> >
> > Dear authors,
> >
> > Thank you for your detailed response and for providing additional experiments.
> >
> > I still have several questions:
> >
> > 1. What are the differences between APPNP and NGC/RNGC? If I understand it correctly, they are equivalent since their graph diffusion process solves the same graph signal denoising problem but with seemingly different solvers. In fact, the Neumann series approximation seems equivalent to the iterative solver in APPNP.
> >
> > 2. Why can NGC/RNGC outperform APPNP in the noising setting?
> >
> > 3. In [1], AirGNN significantly outperforms APPNP in noisy and adversarial feature settings. But it performs the same as APPNP in your results. It seems to contradict the findings in [1]. Could you please provide more details about the experimental setting and hyperparameter tuning?
> >
> > 4. I can understand AGSD might help improve the robustness against graph perturbations when AGSD is used during training, but I do not think it will be helpful in testing/inference. Could you please explain why it can help during testing?
> >
> > [1] Graph Neural Networks with Adaptive Residual.
> >
> > Reviewer ZXuu

---

> > > ### Author Response · Authors · 2022-11-27
> > > **Further response**
> > >
> > > Dear Reviewer ZXuu,
> > >
> > > Thank you for your comments. We address your concerns in the following.
> > >
> > > **Q8:** The difference between APPNP and NGC/RNGC and why NGC/RNGC outperform APPNP in the noising setting.
> > > **A8:** Thanks for your question. First of all, we didn’t claim NGC to be something new. It is a general framework that is derived from the GSD problem. Indeed APPNP is equivalent to NGC if we expand its iterative formulation. There are some minor differences in terms of implementation, e.g., APPNP first uses two MLP layers which include the non-linear activation function to transform the original feature matrix while NGC by default only uses linear layer. For RNGC, as it is derived from the new AGSD problem, it is different from APPNP as we further introduce an additional term $\frac{\varepsilon \mathbf{X} \mathbf{X}^{\top}}{\left\|\mathbf{X} \mathbf{X}^{\top}\right\|_F}$.
> > >
> > > Technically speaking, APPNP is the same method as NGC (but with different hyperparameters) as we mentioned above. Thus we didn’t intend to compare whether NPC is better than APPNP but more like an ablation study (NGC has more layers and larger $\lambda$) to verify our theoretical analysis.
> > >
> > > **Q9:** The experimental setting and hyperparameter tuning?
> > > **A9:** For the baselines, we just follow the experimental setting reported in their paper. For the hyperparameter tuning of NGC, we follow the experimental setting in $S^2$SGC (here following SSGC is not good. maybe just tuned for the best), and we only tune the $\lambda$ and weight decay. The hyperparameters can be found in Appendix I.
> > >
> > > We carefully check the noisy setting in [2] and found out that the settings are not exactly the same as ours. In [2], features of certain nodes are replaced with random Gaussian features, while in our setting, we simply add the Gaussian noise to the current feature matrix. In short, the setting is not the same, which explains the performance difference.
> > >
> > > **Q10:**  Why AGSD help during testing.
> > > **A10:** From Figure 4 and 5, we can know that when testing with no noise, AGSD (RNGC) indeed does not help achieve better performance than GSD (NGC). But when testing with noisy data, AGSD can help as it is trained to be more robust against the perturbations that could lead to small changes in the Laplacian.
> > >
> > > [1] Predict then Propagate: Graph Neural Networks meet Personalized PageRank.
> > >
> > > [2] Graph Neural Networks with Adaptive Residual.
> > >
> > > Best,
> > > Authors

---

> ### Author Response · Authors · 2022-12-09
> **We hope the reviewer find our response helpful**
>
> We appreciate the reviewer for providing many insightful feedbacks, which greatly help us improve the clarity and scope of our paper. We tried to provide a thorough response and additional results. We sincerely hope the reviewer finds our response useful and updates the scores if your concerns have been resolved. We are also open to further discussion if there are further questions.

---

### Official Review · Reviewer_J8PX · 2022-11-03

**Confidence:** 2
**Correctness:** 2
**Technical Novelty And Significance:** 2
**Empirical Novelty And Significance:** 2
**Recommendation:** 6

**Clarity, Quality, Novelty And Reproducibility:**

This paper makes a comprehensive theoretical research and analysis on the time and reason of implicit denoising in GNN, with fair quality.

**Strength And Weaknesses:**

This paper makes a comprehensive theoretical research and analysis on the time and reason of implicit denoising in GNN.

**Summary Of The Paper:**

This paper makes a comprehensive theoretical research and analysis on the time and reason of implicit denoising in GNN, which is an interesting and well written research. Graph Neural Networks (GNN) is widely used in graphic structured data processing because of its powerful representation learning ability. The convergence of noise matrix is studied in this paper. Theoretical analysis shows that implicit denoising depends on connectivity of noise matrix, graph size and GNN structure to a large extent. A robust graph convolution is obtained by solving the signal denoising problem of the extended graph, which improves the smoothness of the NOD representation and the corresponding denoising effect. Extensive empirical evaluation verifies the effectiveness of the proposed model.

**Summary Of The Review:**

I am not currently engaged in research in this field, and I am not very familiar with the research content. AC can make decisions based on the opinions of other reviewers.

---

> ### Author Response · Authors · 2022-11-14
> **Response to Reviewer J8PX**
>
> We thank the reviewer for his/her effort in reviewing our paper.

---

### Author Response · Authors · 2022-11-14
**General response to all the reviews**

First of all, we thank the reviewers for their valuable comments and feedback. In summary, reviewers generally agree that our theoretical analysis of the GNN denoising effect is novel. They appreciate that "the theoretical formulation and analysis are generally solid", "the paper provides a theoretical analysis of the denoising effect of graph convolution", and "the key concepts and intuitions are explained and after a bit of work, are inferred from the paper".

In the following, we address the concerns raised by reviewers.

---

### Decision · Program_Chairs · 2023-01-20

**Decision:**

Reject

**Justification For Why Not Higher Score:**

somewhat limited novelty

**Justification For Why Not Lower Score:**

N/A

**Metareview: Summary, Strengths And Weaknesses:**

The paper studies the phenomenon of implicit denoising in GNNs. The reviewers raised several concerns, among which the fact that the theoretical analysis is based on a simplified GNN architecture that is dissimilar from typically used GNNs, putting doubt in the applicability of the analysis. There was also concern about the discrepancy between theoretical claims and experiments. Some of the doubts were addressed in the rebuttal, but we believe the paper is below the conference bar.